



# A Global Compilation of U-series Dated Fossil Coral Sea-level Indicators for the Last Interglacial Period (MIS 5e)

Peter M. Chutcharavan[1,2], Andrea Dutton[1]

[1]Department of Geoscience, University of Wisconsin – Madison, Madison, Wisconsin, 53706, USA
[2]Department of Geological Sciences, University of Florida, Gainesville, Florida, 32611, USA

*Correspondence to*: Peter M. Chutcharavan (chutcharavan@wisc.edu)

**Abstract.** This dataset is a comprehensive, global compilation of published uranium series (U-series) dated fossil coral records from ~150 – 110 thousand years ago, as well as associated elevation measurements and sample metadata. In total, 1312 U-series measurements from 994 unique coral colonies are included in the current version of the dataset, all of which have been
normalized and recalculated using the same decay constant values. Two example geochemical screening criteria have been included to assist users with identifying altered fossil corals that display geochemical open-system behaviour, and the originally published interpretations on age quality have been preserved within the sample metadata. Additionally, a clear distinction has been made between coral colonies that are in primary growth position, which may be used for relative sea-level reconstructions and colonies that have been transported/reworked, which cannot be used for these purposes. Future research
efforts involving fossil coral sea-level reconstructions should emphasize an "integrated" and holistic approach that combines careful assessment of U-series age quality with high-precision surveying techniques and detailed facies/stratigraphic observations. This database is available at http://doi.org/10.5281/zenodo.4309796 (Chutcharavan and Dutton 2020).

## 1 Introduction and Literature Overview

Uranium-thorium (U-series) dating of Last Interglacial (LIG) fossil corals has long been a key component of the
paleoceanographic toolkit. Early work utilized alpha spectrometry, which has analytical uncertainties on the order of several thousand years for LIG fossil corals. Nonetheless, these early studies provided some of the first radiometric age constraints on the timing of Late Pleistocene glacial-interglacial cycles and were critical for validating the Milankovitch hypothesis (*e.g.,* Broecker et al., 1968; Bender et al., 1979). More recently, the advent of modern mass spectrometric U-series techniques in the mid-1980s reduced analytical uncertainties of LIG fossil coral U-series ages to one thousand years (1 kyr)
or less, allowing workers to precisely determine the timing of the LIG and further refine our understanding of the relationship between orbital forcing, solar insolation, and sea-level/climate change (Chen et al., 1986; Edwards et al., 1987a; Edwards et al., 1987b; Gallup et al., 1994; Stirling et al., 1995; Stirling et al., 1998). In the last three decades, further improvements to existing thermal ionization mass spectrometry (TIMS) methods and the development of robust inductively coupled plasma-mass spectrometry (ICP-MS) techniques have continued to push the boundaries of analytical precision, and





today many labs routinely generate coral U-series ages with an analytical precision of several hundred years for the LIG (*e.g.,* Cheng et al., 2000; Stirling et al., 2001; Andersen et al., 2008; McCulloch and Mortimer, 2008; Cheng et al. 2013).

Global synthesis studies have estimated that the LIG sea-level highstand lasted from approximately 129 to 116 ka and that global mean sea level (GMSL) was likely 6-9 m higher than present (Kopp et al., 2009; Dutton and Lambeck, 2012; Masson-Delmotte et al., 2013; Dutton et al., 2015a). However, the rate, timing and magnitude of GMSL change within the

LIG is still debated, with published interpretations ranging from a single, stable highstand peak to multiple peaks separated by ephemeral sea-level falls (Kopp et al., 2017 and references therein). Reconciling these different interpretations for how sea level evolved during the LIG is crucial for improving our understanding of ice sheet (in)stability during warm periods such as the present Holocene interglacial and for constraining the future sea-level response to human-caused climate change.

Understanding what the global fossil coral record tells us about LIG sea level requires careful interpretations of the age,

elevation, and underlying metadata that comprise a coral relative sea level (RSL) indicator. This is not a trivial undertaking, as data reporting protocols vary by research group and have evolved over the 30+ years that corals have been U-series dated using mass spectrometry (50+ years if alpha spectrometry is considered). It is not only important that the originally published information be collated and reported – it must also be standardized. The dataset should also be easily accessible to users who do not work directly with fossil coral RSL indicators but require a ready-to-use dataset that has already been

quality-checked.

Here we present, to our knowledge, the most comprehensive compilation to date of U-series dated fossil coral RSL indicators for the LIG as a contribution to the World Atlas of Last Interglacial Shorelines (WALIS, https://warmcoasts.eu/world-atlas.html). This work builds upon two previous data compilations (Dutton and Lambeck, 2012; Hibbert et al., 2016) and also includes newly compiled data from several additional studies (Al-Mikhlafi et al., 2018; Bar et

al., 2018; Braithwaite et al., 2004; Dechnik et al., 2017; Kerans et al., 2019; Kindler and Meyer, 2012; Manaa et al., 2016; Muhs et al., 2014; Muhs and Simmons, 2017; Pan et al., 2018; Pedoja et al., 2018; Yehudai et al., 2017).

A sitemap of all the localities included in the database is provided in Fig. 1. The dataset includes 1312 individual U-series measurements and 104 fields for a total of 136,448 entries. All included U-series ages are either (1) dated to between 150 – 110 ka and/or (2) derived from a coral colony that was sampled from an LIG fossil reef unit. U-series ages and isotope

ratios were recalculated using the most recent set of decay constants for $^{234}$U and $^{230}$Th, to conform to data reporting standards that have been established by the U-series community (Cheng et al., 2013; Dutton et al., 2017). Although comprehensive, this dataset is not necessarily exhaustive, and we fully expect that the U-series component of WALIS will expand in the coming years as users continue to add legacy data and data from newly published studies.

We preserved the originally reported values and metadata within WALIS, while also producing two pre-screened,

interpreted versions of the dataset based on data quality that can assist users with identifying fossil coral U-series dates that display open-system behavior. The intention is that this combined approach will ensure that this dataset will adhere to FAIR data principles, being findable, accessible, interoperable and, above all, reusable (Wilkinson et al., 2016). This dataset is open source, and the most recent version can be found at http://doi.org/10.5281/zenodo.4309796 (Chutcharavan and Dutton 2020).

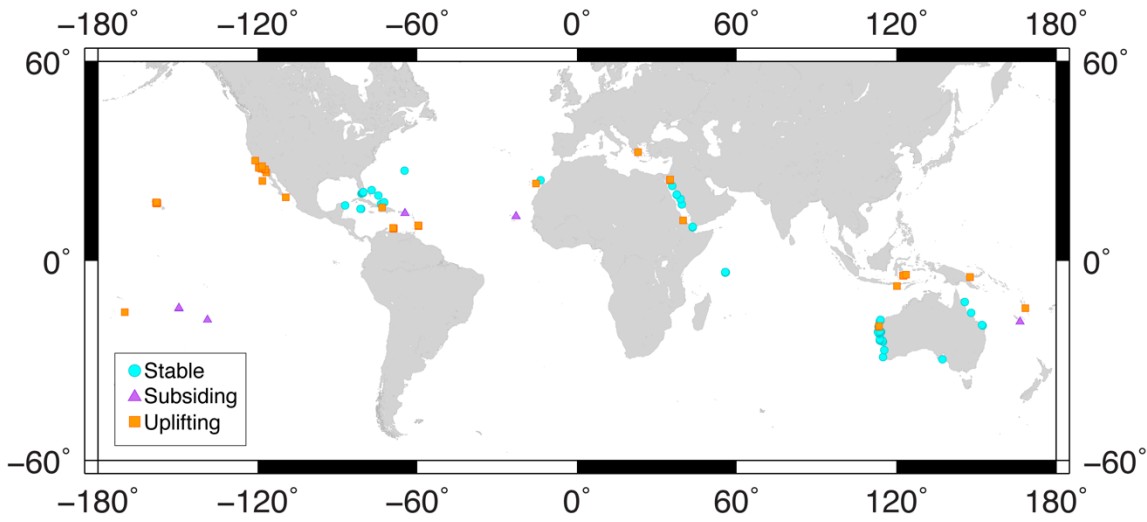

**Figure 1.** Sitemap of U-series dated fossil corals compiled for this study. Sites are differentiated based on regional tectonic setting, with stable sites marked with a cyan circle, subsiding sites with a purple triangle and uplifting sites with an orange square. Map created using GMT v5.4.5 (Wessel et al., 2013).

## 2 Methods

This data compilation is one component of the WALIS project, which seeks to document all previously published geologic and chronostratigraphic constraints on RSL during the LIG. Although the primary focus of our contribution is on the U-series aspect of the fossil coral record, this information is inseparable from the elevation information and associated metadata when reconstructing RSL at fossil reef sites. A U-series-dated fossil coral can be used as an RSL indicator, provided that certain criteria are met. In a recent review, Rovere et al. (2016) proposed that an RSL indicator has three key components:

(1)     The indicator's position, both in terms of geographic coordinates and relative to an established height datum

(2)     The indicator's position relative to local sea level at the time it was deposited and

(3)     Some form of radiometric or chronostratigraphic age constraint on the timing of deposit formation

If a coral with a U-series age has been collected in primary growth position and the vertical position is known based on criteria #1, then the coral is considered an RSL indicator. In this situation, the sample satisfies category #2, as corals are generally considered as marine-limiting because most coral taxa are limited to below mean lower low water/mean low water springs (MLLW/MLWS), although certain coral taxa and growth forms can colonize the intertidal zone. Hence, sea level is considered to have been at or above the elevation of the top of the coral colony. Fossil coral RSL indicators, however, are most useful when the depth the coral was growing at (*i.e.,* paleowater depth, see Section 2.4) is known.



Identification of reliable fossil coral RSL indicators requires careful vetting of each sample's age (*i.e.,* diagenetic screening) and vertical position relative to past sea level. This is important because ignoring additional relevant observations and metadata can result in erroneous conclusions about past sea-level change. In this compilation, we included new paleowater depth interpretations, as well as several screening "scenarios" that were designed to screen out altered samples using a consistent set of defined criteria. These screening scenarios are not intended to be the final word on which coral samples should be accepted/rejected in future studies. Rather, our twin objectives here are to (1) highlight best practices when interpreting fossil coral RSL data, and (2) to provide curated example datasets that are immediately available to WALIS users seek a current best estimate of interpreted RSL in space and time using the coral data. We caution that the screened datasets presented here may not identify every open-system coral, so even U-series ages that pass a particular closed-system criterion still need to be evaluated in the context of existing geologic/sedimentary evidence to assess whether the age is meaningful. In other words, this screening process is only the first step in interpreting the sea level history based on fossil coral data. Additional stratigraphic, sedimentologic, or other metadata may provide justification to modify or reject these preliminary age interpretations. Below, we explain the method we used to develop these datasets and also briefly address the effects of tectonics, glacial isostatic adjustment, and dynamic topography on solid earth displacement, which can cause substantial departures in RSL relative to GMSL.

## 2.1 Database Structure and Major Changes from Previous Compilations

A simplified overview of the WALIS U-series fossil coral dataset and workflow is provided in Figure 2, and the database field descriptors can be found here: https://doi.org/10.5281/zenodo.3961543 (Rovere et al., 2020). Published U-series analyses, elevation measurements and relevant metadata for each dated fossil coral are uploaded into WALIS either (1) manually via an online user interface, or (2) multiple entries are uploaded at once using a spreadsheet template. Once entered into WALIS, all of the uploaded information is added to the WALIS fossil coral U-series database, and each analysis is assigned a unique identifier. Finally, all analyses from corals that can be used as RSL indicators (*i.e.*, that are both in primary growth position and have an associated elevation measurement) are further subset into a fossil coral RSL database. Both databases can then be downloaded by any WALIS user.

This dataset contains several new features that have been added since the Dutton and Lambeck (2012) and Hibbert et al. (2016) compilations. Several key updates include:

(1)     New sample identifiers, which make it easier to identify which U-series analyses are associated with the same coral colony. Sample IDs are reported in the format *XX00-000-000*. The first four digits denote the study that the coral age was published in, whereas the following sets of three numbers represent the coral sample and U-series analysis, respectively. For example, CH91-001-002 is the second U-series age reported for the first coral (here denoted with the number "001"), published in Chen et al. (1991).

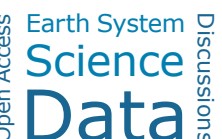

(2)     Sample elevations are now reported both in meters above mean sea level (amsl) and relative to mean lower low water/mean low water springs (mllw/mlws).

(3)     All color coding from the Hibbert et al. (2016) database has been removed. This information is now stored in the "comment" columns.

(4)     The columns for reporting coral taxonomic information have been revamped to allow entry of family, genus, and species information for each coral sample. Coral taxa were updated to reflect the most recent taxonomic classification as reported by the World Register of Marine Species (WoRMS, http://www.marinespecies.org/). Reported coral taxonomic IDs are still preserved, and additional can be added in the comments field for this section.

(5)     All U-series ages from transported corals are now marked as not in primary growth position, even if the original publication explicitly states that the sample is in situ (*e.g.,* an *in situ* clast/conglomerate).

(6)     We have back calculated U-series measurements, when possible, that were not reported in the original publication and had not already been done by Hibbert et al. (2016).

(7)     As with Hibbert et al. (2016), all ages and activity ratios, where appropriate, have been recalculated using the Cheng et al. (2013) decay constants for $^{230}$Th and $^{234}$U. This was done using the open-source software EARTHTIME Redux (ET_redux; https://github.com/CIRDLES/ET_Redux).

(8)     We have restored some of the original information and comments from Dutton and Lambeck (2012) that were not included in the Hibbert et al. (2016) compilation.

(9)     Locality information for Barbados reef terraces have been standardized and reformatted in cases where there were multiple names for the same site.

## 2.2 U-series Diagenetic Considerations

Corals precipitate their skeletons directly from dissolved ions in seawater, forming a calcium carbonate mineral called aragonite. As part of this process, uranium (U) is incorporated at parts per million (ppm) concentrations as impurities within the aragonite crystal lattice, and in ideal, closed-system conditions thorium (Th) concentrations are negligible. This is because of the high particle reactivity of Th, which causes the element to have a relatively short residence time in the water column. Once the coral skeleton has formed, the U-series radiometric clock is effectively started, and the elapsed time is measured by the radioactive decay and ingrowth of $^{230}$Th, $^{234}$U and $^{238}$U as the system returns to secular equilibrium. It is the disequilibrium that arises from the combination of high U concentrations and negligible detrital Th content that enables high-

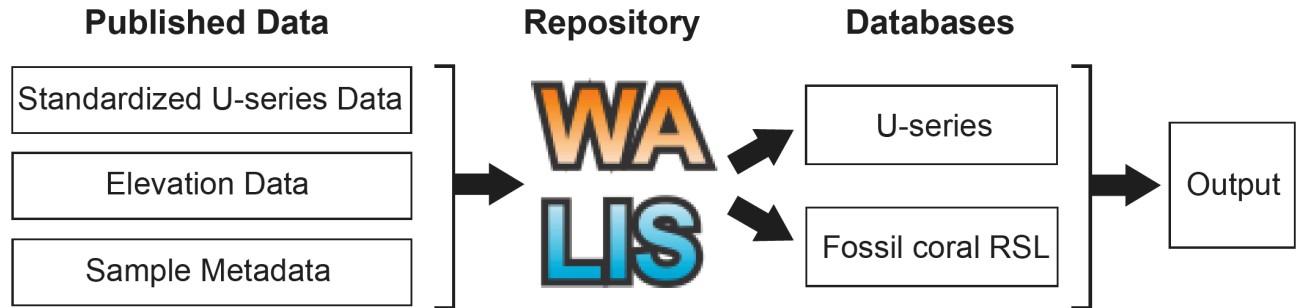

**Figure 2.** Simplified flowchart of WALIS coral U-series database structure. All coral age, elevation and metadata are included
in the "U-series" component of the database, whereas the "fossil coral RSL" database only includes entries from corals that
are in primary growth position.

precision U-series dating of coral skeletal material, thus making fossil corals both valuable RSL indicators and an important
source of absolute age control for other marine-derived sediments (*e.g.,* marine terrace deposits).

Unfortunately, coral skeletal material is also highly susceptible to post-depositional alteration (*i.e.,* diagenesis),
particularly after exposure to meteoric waters, as is often the case with emergent LIG reef units (Thompson et al., 2003). As
a result, a U-series *date* (*i.e.,* calculated from U-series measurements without interpretation) must be carefully evaluated for
signs of geochemical open-system behavior before it can be used to constrain a fossil coral *age*, which is an interpretation of
the U-series date. Prior to U-series dating, coral samples are frequently prescreened using X-ray diffraction (XRD) and thin
section microscopy to identify evidence of recrystallization and/or alteration of coralline aragonite to secondary calcite
minerals. Even coral samples that do not have detectable calcite and are not recrystallized can still yield anomalously
young/old ages for an LIG deposit, indicating that mineralogically pristine samples can still display open-system behavior
with respect to U-series isotopes (*e.g.*, Fig. 3). Therefore, additional geochemical variables are often used to evaluate the
quality of U-series ages.

Several models have been proposed to correct U-series ages that display open-system behavior, but it is well-understood
that patterns of diagenesis in altered corals at a study site follow multiple diagenetic pathways that cannot be explained by a
single model (Henderson and Slowey, 2000; Scholz et al., 2007; Thompson et al., 2003; Villemant and Feuillet, 2003).
While there are circumstances in which altered coral samples may be good candidates for open-system correction, this would
require further analysis of diagenetic trends at each site that is beyond the scope of this study, as no single open-system
model can explain all of diagenetic variability in the dataset. For example, the Thompson et al. (2003) open-system model is
well-suited to correct diagenetic arrays common to Barbados and some localities in Western Australia. It does not, however,
explain all modes of diagenesis present in the fossil coral record (*e.g.*, Fig. 3A; Fig. 4). Hence, this analysis focuses on
assessing closed-system ages.

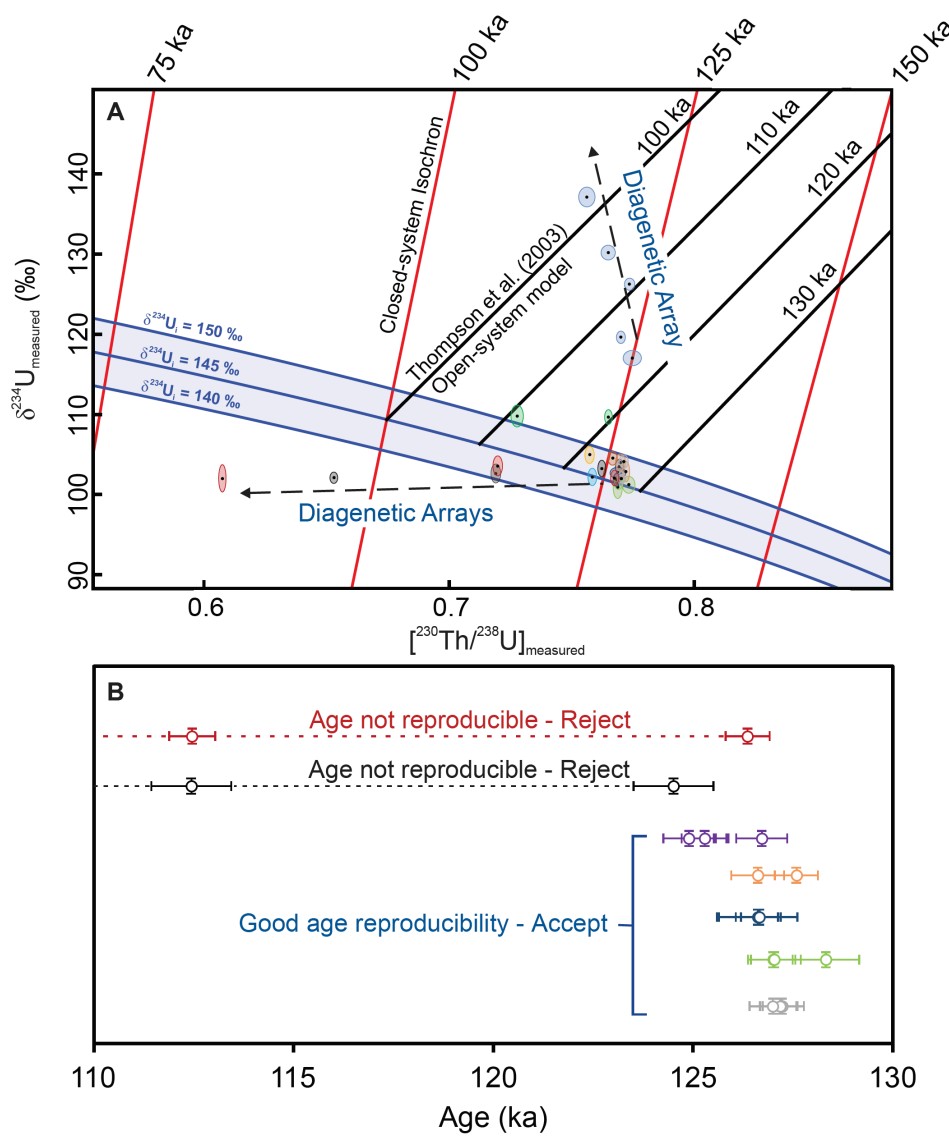

**Figure 3.** Examples of open-system behavior from Last Interglacial fossil coral U-series data. (A) Evolution diagram with
data from the Seychelles plotted. Red lines are closed system isochrons, while black lines are open-system isochrons based on
the Thompson et al. (2003) model. Analyses that fall within the shaded blue region are treated as closed-system ages, assuming
that that the $\delta^{234}U$ value of LIG seawater is the same as today ($\delta^{234}U_{modern}$ = 145 ‰; Anderson et al., 2010; Chutcharavan et
al., 2018). Data points that are the same color represent different subsamples from the same coral colony. Several prominent
diagenetic arrays are indicated with dashed arrows. (B) Analyses from (A) that passed closed-system criteria plotted by age.
While the red and black measurements individually meet closed-system criteria, lack of age reproducibility between different
subsamples from the same coral colony are indicative of open-system behavior, and these ages should be rejected. Data plotted
in (A) and (B) from Dutton et al. (2015b).

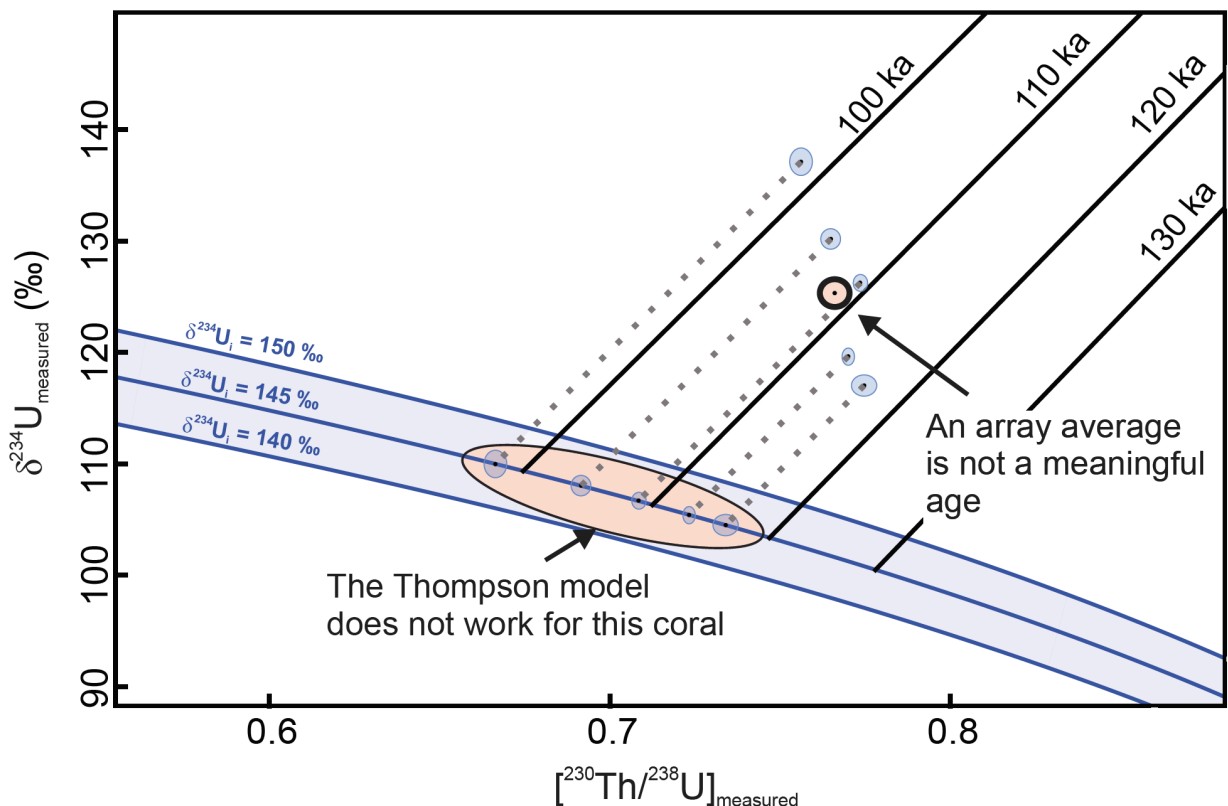

**Figure 4.** Open-system model of Thompson et al. (2003) applied to U-series measurements from Seychelles sample DU15-
180 010 from Fig. 1A (Dutton et al., 2015b). In this case, the diagenetic array is roughly perpendicular to the open-system
isochrons, so the open-system correction does not change the high degree of age variability within this coral colony (~20 ka
in total). For this same reason, an array average does not yield a meaningful age.

**2.3 Geochemical Data Quality Assessment**

Previous studies typically adopted a set of geochemical screening criteria to remove U-series data that have been altered
through open-system behavior (*e.g.,* Scholz and Mangini, 2007). Three of the most common geochemical variables used are:

(1)     Calcite content,

(2)     Detrital $^{232}$Th concentrations, where high $^{232}$Th content can result in anomalously old ages, and

(3)     $\delta^{234}U_i$, which, in a closed-system coral should represent the uranium isotopic composition of ambient seawater at
the time of coral growth.



190 For each published data source, the original list of ages that were accepted/rejected by the study authors is recorded in WALIS. It is often difficult to directly compare screened data between different publications and research groups, as the specific screening criteria applied can vary substantially from study to study. As a result, previous global fossil coral compilations (Dutton and Lambeck, 2012; Hibbert et al., 2016) have applied these screening criteria uniformly across the entire dataset to ensure that only the most geochemically pristine samples were used for sea-level interpretations. Applying a

195 blanket screening criterion, however, results in the vast majority of U-series analyses being rejected and ignores differences that may exist in the nature of diagenesis at different sites and with different coral taxa. Therefore, we applied two sets of screening protocols to the dataset: 1) a "strict" protocol that applies uniform screening cutoffs to each U-series age based on the three geochemical variables listed above, and 2) a "flexible" protocol that allows for site- and sample-specific criteria, particularly where multiple subsamples of the same coral have been dated.

200  The "strict" screening protocol follows the general approach of Dutton and Lambeck (2012) and Hibbert et al. (2016), with some modification in the case where multiple subsamples from a single coral specimen were dated. To be accepted, a sample must have:

(1)  Calcite content < 2 %,

(2)  Detrital $^{232}$Th concentration < 2 parts per billion (ppb), and

205 (3)  $\delta^{234}U_i$ should be within 5 ‰ of the average value for modern corals/seawater (~145 ‰; Andersen et al., 2010; Chutcharavan et al., 2018).

  If any of these values are not reported or cannot be calculated, the U-series age is rejected. Additionally, in the case where multiple subsamples of the same coral pass the strict screening criteria, the ages must also be reproducible (*i.e.,* overlap or nearly overlap within analytical uncertainty), and not lie along a diagenetic array (*e.g.,* Fig. 1). Although this last

210 stipulation regarding age reproducibility is necessary to evaluate corals with multiple dated subsamples properly, it has the consequence of biasing the dataset towards corals that have only been dated once but pass the screening criteria. Ideally, we would only use fossil coral ages that have been reproduced by multiple subsamples should be considered as RSL indicators, to ensure that multiple subsamples from the same coral specimen yield reproducible ages. However, this was not feasible for the dataset considered here, as it would have required rejection of nearly all the coral data that were compiled. Although

215 application of a uniform screening criteria to the global dataset is appealing from a logistical perspective and gives the appearance that data are being treated equally, there can be important methodological differences and additional contextual information that cannot be incorporated using a uniform screening protocol. To address this, we also applied a "flexible" screening protocol that evaluates each study and study site independently, so that nuances in U-series age interpretations could be evaluated.





Many screening decisions are context-based and were addressed separately for each site, but some general modifications to the "strict" screening protocol are addressed here. First, we expanded the calcite screening threshold to include all corals that are below the limit of quantification for the XRD method employed, which can be as high as 4 or 5 % for some studies.

Second, we allowed for a higher $^{232}Th$ threshold of 12 ppb (*i.e.,* a $^{230}Th/^{232}Th$ activity ratio of ~ 500) when age reproducibility can be verified by multiple subsamples from the same coral. This roughly corresponds to a 1 ‰ (or ~0.13

kyr) effect on the measured U-series age, assuming a bulk upper continental crust contaminant (Dutton et al., 2015; Taylor and McLennan, 1995; Wedepohl 1995). Although it has been demonstrated that the composition of detrital thorium contamination can depart from bulk crustal values at different study sites (Cobb et al., 2003; Shen et al., 2008), our approach nonetheless offers a first-order estimate that should approximate the degree of contamination. Additionally, we accepted samples that do not have detrital Th information reported, if rejecting these samples would effectively remove the study site

from the dataset. Cases where this has been done are noted explicitly in the site summaries.

Finally, we expanded the upper limit of the $\delta^{234}U_i$ threshold by 2 ‰, so that the new range of acceptable $\delta^{234}U_i$ values is 140 – 152 ‰, provided that the newly accepted ages are stratigraphically consistent with the other ages from the site. This was done, in part, because the average $\delta^{234}U_{sw}$ value for the LIG is not constrained and there is evidence that the uranium isotopic composition of seawater has varied by several ‰ on glacial – interglacial timescales (Chen et al., 2016;

Chutcharavan et al., 2018). More importantly, it is also clear that there are likely subtle, unresolved biases in interlaboratory calibration protocols that could result in systematic offsets of a few ‰, depending on the lab where a sample was dated (Chutcharavan et al., 2018 and references therein).

The purpose of these screening protocols is, specifically, to identify the highest quality closed-system fossil coral U-series ages that can be used to provide constraints on sea-level change *within* the LIG (*i.e.,* on suborbital/millennial timescales). We

acknowledged that some users may only be interested in knowing whether a geologic feature is broadly constrained to the LIG by the fossil coral U-series data, and we have endeavoured to make such distinctions where applicable in the site descriptions (see Section 3). Users are also cautioned that the screening protocols provided in this manuscript are only intended as guidelines to assist users with identifying coral U-series ages that display closed-system behavior. Just because a U-series measurement fits a set of predetermined geochemical parameters does not automatically imply that an age is robust or that it can provide

meaningful radiometric age constraints on LIG sea-level change. Therefore, it is important for the user to carefully evaluate whether a screened age is consistent with the available geologic context. Additionally, the two example screening protocols provided here are by no means the only way to screen fossil coral U-series data, and we have included a functionality within the WALIS U-series database to upload alternative screening interpretations.



**2.4 Growth Position and Paleowater Depth Uncertainties**

**2.4.1 Identifying growth position corals**

Even if a fossil coral is associated with a robust U-series age, it cannot be treated as an RSL indicator if the vertical position of the sample relative to paleo sea level is not known. This cannot be determined if a coral has been reworked as a cobble or clast since it is not known where the sample originally grew. Therefore, only a fossil coral that has not been transported (*i.e.,* is in primary growth position) can be considered an RSL indicator.

Determining whether a coral sample is in growth position from legacy data can be challenging. The reporting criteria used are not standardized across the literature, and even the terminology used can vary from paper to paper, if it is addressed at all. Generally speaking, the two most common expressions used to indicate that a coral is in place are "growth position" and "*in situ*." Growth position is usually interpreted as expressing greater confidence than *in situ*, as it implies that the coral is in the correct growth orientation or that a clear basal attachment to the reef substrate is visible at the outcrop scale. For the

present study, however, we accepted corals with both designation as an RSL indicator. Hereafter, corals that are listed as either "*in situ*" or "growth position" will be colloquially referred to as in "primary growth position."

There are two unique circumstances for which additional information is required to determine if a coral is in primary growth position. First, some studies refer to a coral specimen as being *in situ*/growth position, yet the depositional context given clearly indicates that the coral has been reworked (*e.g.,* "*in situ* clast" or "*in situ* conglomerate"). We interpreted such

samples as not being in primary growth position. Second, we accepted the designation of "coral framework" as equivalent to *in situ,* and therefore primary growth position, for samples that were collected via drill core (*e.g.,* Camoin et al., 2001; Thomas et al., 2009; Vezina et al., 1999), because in these cases it was impossible to explore the sample's relationship to the rest of the reef unit.

**2.4.2 Constraining paleowater depth uncertainties**

After determining that a coral sample has a reliable U-series age and is in primary growth position, the final challenge involves determining the paleowater depth uncertainty for the coral. As a primary growth position coral, we know that the sample is, at minimum, a marine-limiting RSL indicator, as corals from the highest growth position at an LIG fossil reef site did not necessarily grow directly beneath the paleo sea surface. Many of the studies included in our compilation rely on modern analogue studies of present-day reef ecology to constrain paleowater depth uncertainties.

There are two primary techniques that use the modern analogue approach to constrain paleowater depth (Fig. 5). The first technique is an assemblage-based approach, which examines a series of variables such as coral taxa/growth forms present, associated coralline algal species, and relevant sedimentary context to identify the most probable depth range for the reef unit in which the coral grew (Abbey et al., 2011; Cabioch et al., 1999; Dechnik et al., 2017; Lighty et al., 1982). The assemblage approach is a powerful tool that can substantially reduce the paleowater depth uncertainty for LIG fossil reef

sites. Identifying fossil reef assemblages, however, is by its nature a subjective undertaking. Therefore, users of assemblage-



derived paleowater depth ranges should be aware that these interpretations may change after a study's original publication date, if new stratigraphic context and more robust modern and/or paleoecological studies become available. These are included to help define the paleowater depth uncertainty where possible.

A second approach relies upon modern coral depth distributions to parameterize paleowater depth uncertainty (*e.g.,* Hibbert et al., 2016; OBIS 2014). A significant drawback of using modern depth distributions is that relying upon the full range of growth can greatly overestimate the true depth relative to actual paleo sea-level position, as many corals can grow in a wide range of water depths. For example, individual colonies of the Caribbean coral species *Acropora palmata* have been found growing in water depths up to 22 m in modern reef environments, but this species is more commonly associated with reef crest environments that are < 5 m water depth, with a median depth occurrence of -1.5 m (Lighty et al., 1982;

Hibbert et al., 2016). If field evidence shows that a dated coral was part of a *Acropora palmata* facies where the colonies are in primary growth position, this strengthens the argument that the coral was growing in the <5 m water depth as opposed to closer to the maximum depth range. Therefore, relying on modern coral depth distributions can, in many cases, substantially overestimate the true water depth a fossil coral colony was growing in, and these depth distributions are not a substitute for detailed paleoecological and facies analysis. This is especially true for colonies sampled at or near the highest occurrence of

LIG reef deposits, which were likely growing at the shallower end of their depth ranges.

Whenever possible, we used assemblage-derived paleowater depth estimates, which came either from the original publication or reinterpretations from a subsequent study. If no paleowater depth constraints were available, then we applied the taxon-based modern water depth distributions instead (*i.e.,* the median, upper and lower water depth limits for the 95 % confidence level from Hibbert et al., 2016). All paleowater depth interpretations are current as of the date of publication, but

users are cautioned that some of these interpretations will likely need to be revisited in the future as new studies advance our understanding of LIG and modern reef ecology.

It should also be noted that in the online WALIS database template there are three values that must be given when assigning paleowater depth: 1) estimated paleowater depth and the 2) upper and 3) lower limit of this depth estimate. In the user interface, the upper depth limit is listed first, followed by the estimated paleowater depth and lower limit, with all

depths entered as negative numbers. The estimated paleowater depth does not necessarily have to be the midpoint of the interpreted depth range (*e.g.,* a coral collected from an *Acropora palmata* reef crest facies with an estimated paleowater depth of < 5 m is parameterized as [-5,0,-5], where the first term represents the position below the sea surface and the following two terms represent the uncertainty (+,-). For simplicity's sake, in the main text this will be written as 5 +0/-5 m.

## 2.5 Further Elevation Uncertainty: Causes for RSL departures from GMSL

In general, site-specific fossil reef RSL histories for the LIG diverge from GMSL because of processes such as regional tectonics, glacial isostatic adjustment (GIA), and dynamic topography (Broecker et al., 1968; Farrell and Clark, 1976; Mitrovica and Milne, 2003; McMurtry et al., 2010; Austermann et al., 2017). Although correcting fossil coral RSL records for these processes was not the main focus of this study, it is nonetheless important for a user to be cognizant of this

complication when comparing sea-level records from different sites. It is also worth keeping in mind that although all three

factors affect the uncertainty in the absolute elevation for coral-derived RSL reconstructions, the relative contribution of each

varies from site to site.

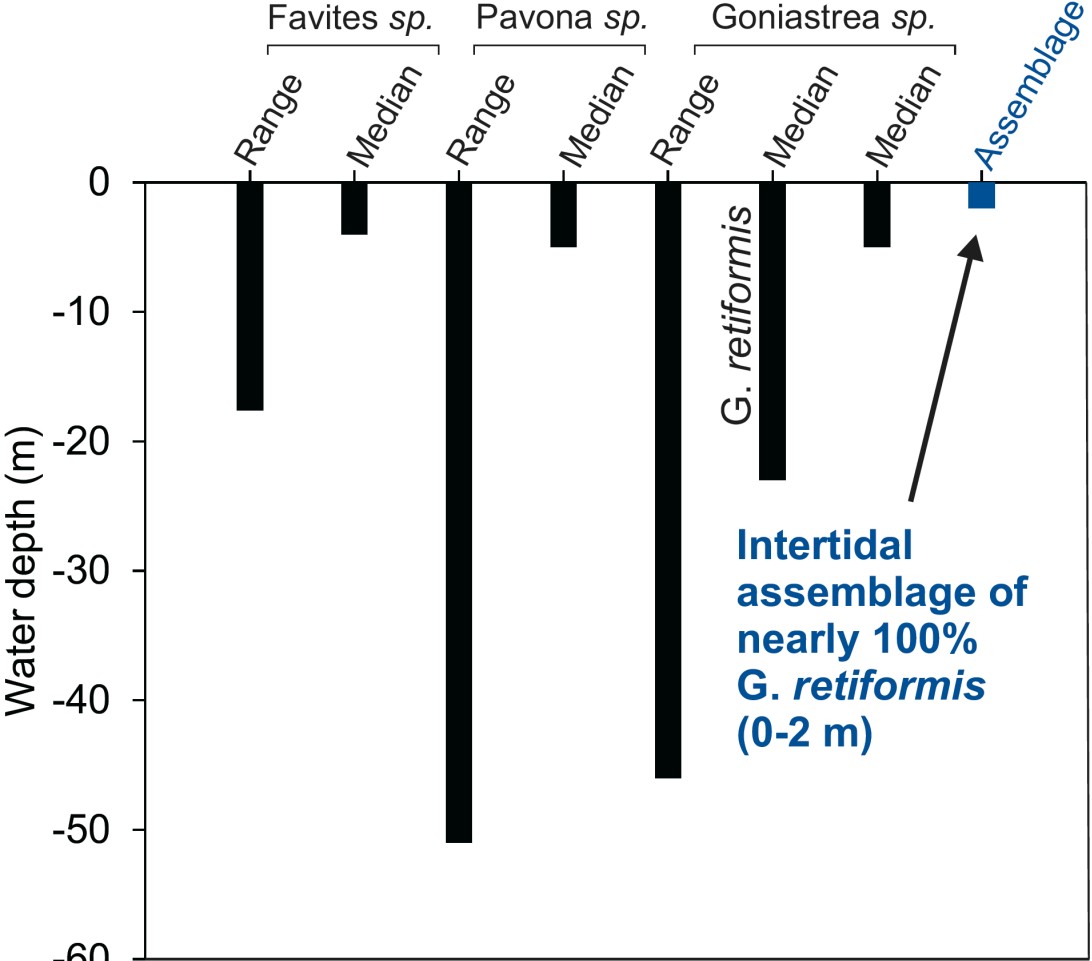

**Figure 5.** Comparison of approaches for interpreting sample paleowater depth based on modern coral depth distributions and reef assemblages for LIG fossil reef outcrops in the Seychelles. Modern depth distributions for the genera Favites, Pavona and

Goniastrea (all of which are found in primary growth position in the Seychelles outcrops) are shown by the black bars in terms of the median and total depth range (95 % confidence; Hibbert et al., 2016). The blue bar is the paleowater depth interpretation for an intertidal assemblage that grew in 0-2 m water depth based on facies interpretations of the fossil reef outcrops and comparison modern reef analogues (Dutton et al., 2015).



### 2.5.1 Tectonic uplift/subsidence

Many of the seminal studies that utilized fossil coral RSL data come from uplifted fossil reef terraces such as those found on the island nation of Barbados and on the Huon Peninsula in Papua New Guinea (Broecker et al., 1968; Bender et al., 1979; Edwards et al., 1987a; Stein et al., 1993). These sites were targeted largely because the uplifted terraces facilitated easy sampling of core material without the need for scientific drilling and because the exposed outcrops enabled detailed facies analysis of the fossil reef morphology and paleoecology. In contrast, some locations in the WALIS database have experienced subsidence since the LIG. In many cases, these sites are located on volcanic hot spot islands, which are subsiding because of crustal loading (*e.g.,* Camoin et al., 2001; Thomas et al., 2009).

The challenge of interpreting RSL records at tectonically active study sites is that the uplift rate must be well-constrained to extract meaningful information about GMSL change. In many cases, previous workers estimated uplift/subsidence rates using the highest growth position coral from an LIG unit (*e.g.,* McMurtry et al., 2010). The general formula used to correct for tectonic activity is:

$$E_{\text{corrected}} = E_{\text{measured}} - rt \tag{4-1}$$

where $E_{corrected}$ is the subsidence-corrected elevation, $E_{measured}$ is the present-day elevation, $r$ is rate of elevation change (positive if uplifted, negative if subsiding) and $t$ is the age of the sample. The rate of uplift/subsidence is itself determined by:

$$r = \frac{(E_{max} - E_{\text{LIG}})}{t_{LIG}} \tag{4-2}$$

where $E_{max}$ is the elevation of the highest growth position coral (regardless of whether the coral has a U-series age), $E_{LIG}$ is the peak elevation of the LIG highstand based on GMSL, and $t_{LIG}$ is the timing of the LIG highstand. This approach, however, does not yield the true uplift/subsidence rate attributable to local tectonics and/or volcanic loading. This is because (1) there is still considerable uncertainty surrounding the actual $E_{LIG}$ value and (2) even if this value was well-constrained, both local RSL and the timing of LIG highstand would still often depart from $E_{LIG}$ and $t_{LIG}$ because GIA effects (Creveling et al., 2015). Therefore, uplift/subsidence rates in this study are included for conceptual purposes only, as uplift/subsidence corrected coral elevations do not typically provide precise absolute elevation constraints for the position of past sea level.

Values used for $E_{LIG}$ and $t_{LIG}$ vary from study to study, so we standardized the dataset by recalculating $r$ and $E_{corrected}$ for tectonically active sites using a value of 7.5 ± 1.5 m for $E_{LIG}$ and 123 ± 6 ka for $t_{LIG}$ based on the midpoint of LIG timing and magnitude given by Dutton et al. (2015a). Interpreted uplift/subsidence are not prescriptive, and these corrections should be reevaluated as new information becomes available.

### 2.5.2 Glacial isostatic adjustment

The advance and retreat of large continental ice sheets during the last glacial cycle caused long-lasting, global perturbations to the Earth's gravity field and rotation that persist to this day (Farrell and Clark, 1976; Mitrovica and Milne, 2003). This phenomenon, called glacial isostatic adjustment or GIA can cause meter-scale changes in RSL at fossil reef sites that must be





addressed to extract meaningful GMSL information for fossil coral sea-level indicators (Dutton et al., 2015a). Indeed, the 6-9 m estimate for the peak magnitude of the LIG highstand has been inferred from global compilations of RSL records that were corrected for GIA effects (Dutton and Lambeck, 2012; Kopp et al., 2009).

     The magnitude of the difference between RSL and GMSL at fossil reef sites is spatially variable, depending in part on the proximity to past continental ice sheets. For instance, GIA modelling predicts a gradient in RSL across the circum-
Caribbean region, as many of the sites were sitting atop or proximal to the peripheral bulge of the Marine Isotope Stage (MIS) 6 ice sheet that covered North America (Dutton and Lambeck, 2012). This is supported by recent field surveys from the Bahamas, which revealed a ~1 m difference between the highest primary growth position corals from LIG deposits on San Salvador and Great Inagua Island (Skrivanek et al., 2018). In contrast, so-called "far-field" sites such as the Seychelles and Western Australia are located much farther from continental ice sheets, which reduces the influence of GIA and thus
enables them to more closely track the magnitude of GMSL change during the LIG (*e.g.,* Dutton et al., 2015b; O'Leary et al., 2013; Stirling et al., 1998). This remains an active area of research, particularly for constraining the global extent and retreat of MIS 6 (~136 – 129 ka) continental ice sheets (Clark et al., 2020; Creveling et al., 2015; Dendy et al., 2017; Hay et al., 2014).

### 2.5.3 Dynamic topography

Dynamic topography is vertical displacement of the solid Earth caused by mantle convection. Previous work demonstrated that the effect of dynamic topography on million-year timescales is of a similar order of magnitude to apparent changes in GMSL inferred from RSL records (Moucha et al., 2008; Müller et al., 2008; Rowley et al., 2013). Recent work demonstrated that this is also the case for the LIG, in that DT can cause meter-scale differences in RSL between the LIG and the present day at some localities (Austermann et al., 2017). These studies clearly demonstrate that the effect of DT on LIG RSL records
is nontrivial, and further work is needed to assess how mantle dynamic topography may affects interpretations of past sea level from fossil reef sites.

     In summary, there are both local (tectonic) and global-scale (GIA, DT) processes that can cause RSL at a fossil reef site to depart from the global mean, and they must be accounted for to extract a robust GSML signal using U-series ages and elevations of fossil corals. Although GIA and DT influence interpretation of RSL compared to GMSL, we do not provide those
interpretations here. Instead, this study was undertaken to define RSL at each site so that robust RSL interpretations are available that can be used to constrain such processes and, by extension, GMSL.

### 3 Last Interglacial Fossil Coral Database

An overview of the coral U-series ages available in the dataset is included below, organized alphabetically by geographic study area. Each entry, where appropriate, contains the following:
(1)       The total number of U-series ages available for the study area and the number of unique coral specimens dated;



(2)    If any of the corals were dated in duplicate, triplicate, etc.

(3)    How many ages were accepted by the original publication

(4)    How many ages (if any) pass the flexible and strict screening protocols,

(5)    Identification of corals that pass screening and are in primary growth position,

(6)    Mention of previous interpretations of paleowater depth uncertainty and what water depth uncertainties were assigned by the present study, and

(7)    If the site is tectonically uplifting, subsiding, or stable

A summary of the coral U-series ages that passed the strict and flexible screening protocol is provided in Supplement S1, and the "flexible" protocol is also coded into WALIS as the preferred screening protocol utilized in this manuscript.

In total, 141 U-series ages were accepted from 104 unique coral samples passed the strict screening protocol, whereas 286 ages from 215 samples were accepted under the flexible protocol (Table 1). We did not include coral U-series ages that were measured using the considerably less precise dating method of alpha spectrometry, but the ability to add alpha dates is present in the WALIS user interface. The addition of alpha spectrometry ages to this dataset by community members is encouraged, especially for sites where mass spectrometric U-series measurements are not available.

**3.1 The Bahamas**

Corals were U-series dated from emergent LIG reef deposits on Great Inagua, San Salvador, and Abaco Islands in the Bahamas (Chen et al., 1991; Hearty et al., 2007; Thompson et al., 2011). A total of 200 U-series ages from 142 unique coral specimens were reported, with 29 of these corals dated at least in duplicate. In total, the original study authors accepted 49 U-series ages from 37 coral samples as closed-system ages. Thompson et al. (2011) did not use closed-system ages and

instead applied an open-system correction to each sample. Under the strict screening protocol, 35 U-series ages from 26 coral samples were accepted. This number increased to 43 U-series ages from 29 corals when the flexible screening protocol was applied. Sample ages that passed flexible screening ranged from $131.3 \pm 1.4$ ka for CH91-002 to a weighted mean age of $119.8 \pm 0.3$ ka for TH11-023 (weighted mean ages are reported where multiple subsamples of the same coral passed the screening criteria).

Several site-specific adjustments were made under the flexible screening protocol. First, the $^{232}$Th concentrations for Chen et al. (1991) were recalculated using the published $^{230}$Th/$^{232}$Th activity ratios from the manuscript supplement, in certain cases for which only one $^{232}$Th concentration was reported for multiple subsamples of the same coral. Second, we only considered samples that were dated at least in duplicate from Thompson et al. (2011), as calcite content was not reported in that study and there are no elevation data from which stratigraphic relationships can be derived. Finally, we





**Table 1.** Summary of samples that passed closed-system screening protocols.

| Location | Total[a] Analyses | Samples | Published Analyses | Samples | Strict (this study) Analyses | Samples | Flexible (this study) Analyses | Samples |
|---|---|---|---|---|---|---|---|---|
| Bahamas | 200 | 142 | 49 | 37 | 35 | 26 | 43 | 29 |
| Baja California | 26 | 26 | 16 | 16 | 0 | 0 | 0 | 0 |
| Barbados | 141 | 107 | 40 | 33 | 24 | 17 | 41 | 28 |
| Bermuda | 9 | 9 | 7 | 7 | 2 | 2 | 3 | 3 |
| California | 153 | 148 | 34 | 32 | 0 | 0 | 4 | 3 |
| Canary Islands | 2 | 2 | 2 | 2 | 0 | 0 | 0 | 0 |
| Cape Verde | 10 | 6 | 10 | 6 | 0 | 0 | 1 | 1 |
| Curacao | 25 | 5 | 5 | 4 | 1 | 1 | 4 | 3 |
| Eritrea | 7 | 6 | 6 | 5 | 0 | 0 | 0 | 0 |
| Florida | 55 | 51 | 15 | 13 | 4 | 3 | 13 | 10 |
| French Polynesia | 19 | 12 | 6 | 3 | 5 | 3 | 9 | 5 |
| Grand Cayman | 15 | 15 | 12 | 12 | 0 | 0 | 0 | 0 |
| Great Barrier Reef | 40 | 14 | 11 | 5 | 3 | 2 | 7 | 3 |
| Greece | 2 | 2 | 2 | 2 | 0 | 0 | 0 | 0 |
| Gulf of Aqaba | 67 | 18 | 6 | 4 | 2 | 2 | 2 | 2 |
| Haiti | 3 | 2 | 3 | 2 | 0 | 0 | 3 | 2 |
| Hawaii | 82 | 72 | 59 | 52 | 25 | 23 | 34 | 29 |
| Indonesia | 21 | 21 | 14 | 14 | 4 | 4 | 10 | 10 |
| New Caledonia | 19 | 15 | 0 | 0 | 0 | 0 | 0 | 0 |
| Niue | 1 | 1 | 1 | 1 | 0 | 0 | 0 | 0 |
| Papua New Guinea | 47 | 34 | 11 | 7 | 11 | 5 | 13 | 7 |
| Saudi Arabia | 25 | 25 | 17 | 17 | 1 | 1 | 2 | 2 |
| Seychelles | 67 | 31 | 38 | 24 | 5 | 3 | 25 | 14 |
| Southern Australia | 4 | 4 | 1 | 1 | 0 | 0 | 0 | 0 |
| St. Croix, USVI | 6 | 6 | 5 | 5 | 4 | 4 | 5 | 5 |
| Turks and Caicos | 19 | 19 | 13 | 13 | 0 | 0 | 0 | 0 |
| Vanuatu | 3 | 2 | 3 | 2 | 3 | 2 | 3 | 2 |
| Western Australia | 176 | 156 | 59 | 55 | 9 | 5 | 61 | 56 |
| Yemen | 35 | 33 | 0 | 0 | 0 | 0 | 0 | 0 |
| Yucatan | 33 | 10 | 7 | 5 | 3 | 1 | 3 | 1 |
| **Total** | **1312** | **994** | **452** | **379** | **141** | **104** | **286** | **215** |

[a]Includes all reported analyses and samples (as opposed to the other three pairs of columns, which only include analyses that were accepted by the respective screening protocol).

accepted ages from sample CH91-023 as closed-system since the ages were reproducible and calcite content was on the

cutoff threshold at 2 %.



Assessing whether corals from the Bahamas dataset were in primary growth position is challenging. Chen et al. (1991)
applied the term "*in situ*" to describe both growth position corals that are part of the reef framework and detrital coral rubble
that had been cemented in place. For the present compilation, we categorized all corals derived from rubble layers as "not *in
situ*/growth position." Using this approach, a total of 14 ages from Chen et al. (1991), derived from 11 coral specimens, can
be treated as RSL indicators under the flexible screening protocol. Previous workers assigned a paleowater depth range of 3

to 4 m for the Cockburn Town and Devil's Point sites (Chen et al., 1991). A more recent study, however, reevaluated the
vertical position and facies characteristics of the two fossil coral reefs using high-precision surveying techniques, and
published new paleowater depth interpretations (Skrivanek et al., 2018). It is difficult to compare the present dataset to the
reef zones described in Skrivanek et al. (2018), as Chen et al. (1991) did not distinguish between reef units in their study. All
of the corals in primary growth position, however, were colonies of *Pseudodiploria clivosa* or *Orbicella annularis*, which

were found in units with interpreted paleowater depths of 0.2 – 5 m at Devil's Point reef and 0.2 – 3 m at Cockburn Town
(Skrivanek et al., 2018).

Thompson et al. (2011) distinguished corals that were derived from a rubble layer from those collected from *in situ* reef
framework but gave no elevation information associated with each sample, so none of the samples are used as RSL
indicators here. Though elevation estimates were provided in Thompson et al. (2011) for each reef unit, these elevations do

not always match those subsequently surveyed at the same sites, calling into question the use of those approximate
elevations (Skrivanek et al., 2018). Primary growth position corals can, however, still be used to constrain the maximum age
of each fossil reef, even without published elevation data. A total of 5 corals (11 analyses total) from Thompson et al. (2011)
are in primary growth position and passed the flexible screening criteria. These ages range from 127.3 ± 0.6 ka to 119.8 ± 0.3
ka for the Devil's Point reef, and 125.2 ± 1.5 ka to 122.2 ±1.7 ka for the Cockburn Town reef.

**3.2 Baja California, Mexico**

U-series coral ages were reported for three locations along the Pacific coast of Baja California, Mexico (Muhs et al., 2002a).
Corals collected for that study came from detrital sedimentary deposits on marine terraces and, therefore were not in primary
growth position and cannot be used as strictly reliable RSL indicators. Instead, the study authors used coral U-series ages as
a constraint on the maximum age of the terraces. In total 26 corals were dated, and the study authors accepted 16 of the U-

series ages. None of these ages passed the strict or flexible closed-system criteria.

**3.3 Barbados**

Barbados is one of the most intensely studied LIG fossil reef localities in the world, with 141 U-series analyses reported for
107 corals from 11 separate studies (Bard et al., 1990; Blanchon and Eisenhauer, 2001; Cutler et al., 2003; Edwards, 1997;
Edwards et al., 1987b; Gallup et al., 1994, 2002; Hamelin et al., 1991; Muhs and Simmons, 2017; Speed and Cheng, 2004;

Thompson et al., 2003). The island is located on an accretionary prism and has experienced differential uplift since the LIG.
Local uplift rates vary from ~0.2 m/kyr in the north and south of Barbados to as high as ~0.5 m/kyr near the Clermont





Nose/University of the West Indies transect near the middle of the island (*e.g.,* Muhs and Simmons, 2017; Taylor and Mann, 1991), so care must be taken when applying subsidence corrections to the Barbados dataset. Additionally, the dataset can be challenging to interpret, as there are multiple names for some localities, and some coral samples have been dated in two or

more studies. To facilitate data accessibility, we standardized site location names (*e.g.,* all LIG samples from the Clermont Nose area were given the site name "Univ. West Indies (UWI) Transect"), and we endeavored to link U-series measurements across multiple studies that were derived from the same coral colony.

Of the U-series ages reported, the original study authors accepted 40 ages from 33 unique coral specimens. It should be noted that Thompson et al. (2003) did not apply closed-system screening criteria; rather, an open-system model was used.

Under the strict screening protocol, a total of 24 U-series ages were accepted from 17 corals, whereas the flexible protocol accepted 41 ages from 28 corals. Ages from the flexible screening protocol range from $103.8 \pm 1.0$ ka (BL01-001-001) to $172.5 \pm 1.4$ ka (GA02-006-001). The oldest age was sampled from "Lazaretto unit," which is part of the LIG Rendezvous Hill terrace, but this unit is actually MIS-6 in age (Speed and Cheng, 2004). It should be noted that one sample which passed both screening protocols, GA02-032-001 ($136.4 \pm 0.9$ ka) was rejected, as this age was eventually retracted by the Gallup et

al. (2002) study authors after multiple dated subsamples from the same colony were unable to reproduce the reported age. Several of the corals were also dated using Pa-Th methods (Cutler et al., 2003; Edwards, 1997; Gallup et al., 2002).

Of the samples that passed flexible screening, a total of 9 U-series ages (7 corals total) came from corals that were stated as being in primary growth position, with ages ranging from $103.8 \pm 1.0$ ka (BL01-001) to $172.5 \pm 1.4$ ka (GA02-006). The number of RSL data points is increased to 30 ages from 22 corals by including all samples that were not explicitly identified

as transported clasts or cobbles. In cases where primary growth position corals are derived from a reef crest facies, we assigned a paleowater depth of < 5 m, which is the typical depth range for modern Caribbean reef crest environments (Lighty et al., 1982). In all other cases, we applied the taxon-derived modern depth distributions of (Hibbert et al., 2016).

### 3.4 Bermuda

LIG-aged corals are present at Grape Bay on the southern side of Bermuda (Ludwig et al., 1996; Muhs et al., 2002b). These

deposits are inferred to be originally derived from patch reefs, but are not in primary growth position and may have been storm-derived (Muhs et al., 2002b). Therefore, the ages presented in these studies represent a constraint on the maximum age of the rubble deposit but cannot be used as RSL indicators. In total, nine corals were dated from Grape Bay, and the authors originally accepted seven of the ages. Only two corals pass the strict closed-system criteria: MU02-019, with an age of $116.9 \pm 0.9$ ka and MU02-020, with an age of $113.7 \pm 0.9$ ka. A third coral, MU02-026 ($118.7 \pm 0.9$ ka) is also accepted once the

flexible protocol has been applied.

### 3.5 California, United States

Several studies reported U-series coral ages from marine deposits along the southwest coast of California and several of the Channel Islands (Muhs et al., 2012, 2006, 2002). These samples are solitary *Balanophyllia elegans* corals from detrital



sedimentary deposits on marine terraces and are therefore not in primary growth position and cannot be used as RSL
indicators. Instead, the study authors used the coral U-series ages as a constraint on the maximum age of terrace formation.
A total of 153 U-series ages were reported for 148 unique coral specimens, with four of the corals dated in duplicate. The
study authors accepted 34 of the ages (32 corals in total). None of the ages passed the strict closed-system criteria, but four
ages from three corals were accepted under the flexible protocol: MH02-075 (118.3 ± 0.6 ka), MH02-077 (119.9 ± 0.8 ka)
and MH06-013 (weighted mean age of 118.8 ± 0.7 ka).

### 3.6 Canary Islands

Two Last Interglacial ages are reported from Gran Canaria and Lanzarote in the Canary Islands (Muhs et al., 2014). Both
corals were *Siderastraea radians* fragments collected from marine deposits and were used to determine the maximum age of
the deposits and constrain local uplift rates. The authors accepted both ages, assigned a 0.017-0.050 m/kyr uplift rate for the
Gran Canaria site, and determined that the Lanzarote site had not been subjected to significant uplift since the LIG. Neither
age passed the strict nor the flexible screening protocols, but they do broadly constrain the age of their respective deposits to
the LIG.

### 3.7 Cape Verde

Zazo et al. (2007) reported U-series coral ages that were used to constrain the age of marine terrace conglomerates on Sal
Island, Cape Verde. In total, 10 U-series ages were reported for five corals (and one hydrozoan), with one coral sample
(ZA07-004) analyzed five times. All coral ages were accepted by the study authors, but only one age from coral ZA07-004
(129.5 ± 4.0 ka) passed the flexible protocol. This coral is not in primary growth position and cannot be used to constrain
RSL.

### 3.8 Curaçao

U-series ages have been reported for multiple outcrops of the LIG Hato unit on the island of Curaçao, for a total of 25 ages
from 15 unique coral colonies (Hamelin et al., 1991; Muhs et al., 2012b). Curacao is slowly uplifting, with an estimated
uplift rate of 0.026 to 0.054 m/kyr, based on the "highest inner edge" elevation of the Hato unit at 12.4 m (Muhs et al.,
2012b). In total the study authors accepted five U-series ages from four unique coral specimens. Under the strict screening
protocol, this is reduced to a single age of 118.8 ± 0.8 ka from sample SC78-005-002. The flexible protocol adds three
additional ages: two from MU12-001, with a weighted mean age of 126.6 ± 0.7 ka, and an age of 118.7 ± 1.2 ka from coral
SC78-002-002.

All samples which passed the flexible screening criteria were in primary growth position. Based on the
paleoenvironmental interpretations of Muhs et al. (2012b), samples SC78-005 and MU12-001 were part of an *Acropora
palmata* dominated reef crest facies growing in 0 – 5 m water depth, which we adopted for this study. Paleoenvironmental



interpretations and stratigraphic context were not provided for sample SC78-002. Therefore, the modern depth distribution

for *Diploria* sp., from Hibbert et al. (2016) was applied in this case.

### 3.9 Eritrea

Fossil corals of Last interglacial age were reported for the Abdur Reef Limestone on the Red Sea coast of Eritrea (Walter et al., 2000). The Eritrean coast is tectonically active and is estimated to be uplifting by 0.06 m/kyr based on the elevation of the LIG reef deposits (Hibbert et al., 2016). In total, seven U-series ages were reported for 6 corals, with one coral dated in

duplicate. In the original study, the authors accepted all ages, except for one from coral WA00-006, which had anomalously low U content and an age that was older than expected. None of the ages passed the strict nor the flexible closed-system screening criteria.

### 3.10 Florida, United States

Fossil corals have been dated from multiple sites across the Florida Keys (Fruijtier et al., 2000; Muhs et al., 2011; Multer et

al., 2002). In total, 55 U-series ages were reported for 51 unique coral samples, with 4 corals dated in duplicate. In total, the study authors accepted 15 of the ages from 13 coral samples. Under the strict screening criteria, 4 ages were accepted from 3 unique coral specimens from Windley Key: MU11-026, with a weighted mean age of $115.1 \pm 0.6$ ka; MU11-034, with an age of $114.1 \pm 0.6$ ka; and MU11-037, with an age of $120.4 \pm 0.8$ ka. Using the flexible screening protocol, the total number of analyses accepted increases to 13, from total of 10 unique coral specimens ranging from a weighted-mean age of $123.0 \pm$

$0.6$ ka (MU11-012) to a weighted-mean age of $113.7 \pm 0.6$ ka (MU11-034). One of these samples, MU11-005, was accepted despite failing the strict $^{232}$Th criterion, as it was only marginally higher (2.4 ppb) and passed both the calcite and $\delta^{234}U_i$ thresholds. It should be noted that samples from Muhs et al. (2011) and Multer et al. (2002) appeared to have a 5 % limit of quantification for their XRD techniques, so all samples from these studies were interpreted to have acceptable calcite content.

All 10 corals that passed the flexible screening protocol were in primary growth position and, therefore, can be used as RSL indicators. These samples were collected from outcrops of the Key Largo Limestone at Windley Key and Key Largo, with sample elevations ranging from 2-5 m above mean sea level (amsl) (Muhs et al., 2011). The dominant coral taxa in the outcrops studied at both localities were massive *Orbicella annularis* and *Pseudodiploria strigosa*, with the Windley Key site also containing *Copophyllia natans*. Several estimates of paleowater depth for the Key Largo Limestone have been

published and range from < 3 m to as much as 12 m water depth (Fruijtier et al., 2000; Muhs et al., 2011; Perkins, 1977; Stanley, 1966). Most recently, Muhs et al. (2011) interpreted this facies to have a minimum water depth of 3 m based on the optimal depth range for these three coral species from a modern ecological survey of reefs in the Florida Keys and Dry Tortugas (Shinn et al., 1989).

        We adopted the 3 m estimate of Muhs et al. (2011) as the most probable paleowater depth for the LIG deposits at Key

Largo and Windley Key and further parameterized the possible range of paleowater depths. As stated by Muhs et al. (2011),



the optimum water depths for *Pseudodiploria strigosa* and *Copophyllia natans* are 3-10 m and 2-10 m, respectively, whereas the optimal depth range for *Orbicella annularis* is substantially wider, at 3-45 m (Shinn et al., 1989). The global compilation of modern coral data from Hibbert et al. (2016), however, indicates that the modern *Montastrea annularis* distribution has a median depth closer to 10 m (upper 95 % confidence interval = 17 m water depth). Given this additional information, we
parameterized the paleowater depth uncertainty for these samples as 3 +7/-0 m.

## 3.11 French Polynesia

Many of the islands and archipelagos in French Polynesia are former hot spot volcanoes that are subsiding because of volcanic loading. In these cases, LIG deposits are often located below sea level and can only be accessed via scientific drilling. U-series ages from corals have been published from two locations in French Polynesia: Mururoa Atoll in the
Tuamotu Archipelago and offshore drilling at Tahiti during IODP Expedition 310 (Camoin et al., 2001; Thomas et al., 2009b). Both Mururoa Atoll and Tahiti have been subject to subsidence since at least the late Pleistocene. The subsidence rate at Mururoa was estimated to be ~0.07 – 0.08 m/kyr using K-Ar dating of the volcanic basement and on the location of the LIG unit 3 m below the modern reef (Trichet et al., 1984; Camoin et al., 2001). At Tahiti, the subsidence rate is an order of magnitude greater and is commonly estimated to be 0.25 m/kyr with a total possible range of 0.2-0.4 m/kyr (Le Roy 1994;
Bard et al., 1996; Thomas et al., 2012). To date, no coral of Last Interglacial age have been discovered at Tahiti, but several corals from the IODP record have been dated to late MIS 6 (Thomas et al., 2009).

The existing dataset for French Polynesia contains 19 U-series dates from 12 corals, with 5 corals from Tahiti dated in duplicate and one in triplicate. Of the six corals analyzed from Mururoa, only one age (CA01-007) was accepted by the study authors. This sample also passed the strict screening protocol and has a recalculated age of 126.0 ± 2.2 ka. For the samples
from Tahiti, two corals (TH09-001 and TH09-003) were accepted by the study authors. Based on the strict screening protocol, dates from two corals and four unique U-series measurements passed screening: TH09-003, with a weighted-mean age of 133.9 ± 0.4 ka and TH09-008, with a single age of 134.0 ± 0.4 ka. By employing the flexible screening protocol, four additional U-series ages can be included from two corals: TH09-001 (weighted mean age of 138.0 ± 0.5 ka) and TH09-005 (weighted mean age of 137.0 ± 0.5 ka).
Sample CA01-007 from Mururoa was reported as being "reworked" (Camoin et al., 2001). Samples TH09-001, TH09-003, TH09-005 and TH09-008 are both interpreted as being in growth position, and thus can be used as RSL indicators. TH09-003 an TH09-008 are massive *Porites* sp., which is commonly associated with depth ranges of 0-25 m, while TH09-001 and TH09-005 were associated with a shallower facies interpreted as growing in 0-6 m water depth (Thomas et al., 2009).




### 3.12 Grand Cayman, Cayman Islands

Fossil corals have been dated from multiple localities across Grand Cayman (Coyne et al., 2007; Vezina et al., 1999). In total, 15 corals from the Last Interglacial and late MIS-6 were dated, with the authors accepting all but three ages. All of the ages were rejected by the strict and flexible closed-system screening protocols.

### 3.13 The Great Barrier Reef, Australia

Two studies have reported LIG U-series ages from the Great Barrier Reef, which were collected via scientific drilling on modern reef flats (Braithwaite et al., 2004; Dechnik et al., 2017). A total of 40 ages from 14 unique coral specimens were reported, and the authors originally accepted 11 of the ages from five corals. All but one of these corals (BR04-001) were dated in triplicate. Under the strict screening protocol, three ages were accepted from two coral samples (DE17-001, DE17-003). When the flexible protocol was applied, the total number of accepted ages expanded to include seven ages from three corals: DE17-001, with a weighted mean age of 128.7 ± 0.7 ka; a single age from DE17-003, which was dated to 126.1 ± 0.5 ka; and DE17-004, with a weighted mean age of 127.7 ± 0.5 ka. Coral DE17-004 had higher calcite content (6.5 %) but was nevertheless accepted because the ages were stratigraphically consistent with DE17-001/003 and the sample passed the $^{232}$Th and $\delta^{234}$U$_i$ thresholds.

All three corals that passed the flexible screening protocol were in primary growth position and can be used as RSL indicators. Using the coralgal assemblage interpretations, Dechnik et al. (2017) assigned these samples a 0-6 m paleowater depth range that we also adopted.

### 3.14 Greece

One study reported corals with Last Interglacial ages from uplifted terraces on the Perakora Peninsula, Greece, which were originally used to constrain local uplift rates (Dia et al., 1997). The authors reported that the two corals with Last Interglacial ages (DI97-002, DI97-003) showed signs of open-system behavior, based on uranium isotopes ($\delta^{234}$U$_i$ > 200 ‰), high detrital $^{232}$Th concentrations (~ 7 ppb) and anomalously low $^{87}$Sr/$^{86}$Sr ratios. DI97-002 and DI97-003 did not pass the strict nor the flexible closed-system screening criteria.

### 3.15 Gulf of Aqaba

Three studies reported U-series ages on corals from uplifted terraces along the Gulf of Aqaba (Bar et al., 2018; Manaa et al., 2016; Yehudai et al., 2017). Bar et al. (2018) inferred an uplift rate 0.13 m/kyr for the northeastern Gulf of Aqaba based on the present-day elevation of the coral terraces and the timing of diagenesis for altered fossil corals dated by Yehudai et al. (2017). In total, 67 U-series ages were reported for 18 unique coral specimens, with the majority of samples dated in triplicate or greater. In total, the study authors accepted six of these ages from four coral samples. Under both the strict and



flexible closed-system screening criteria, only two samples were accepted from the Upper Haql Terrace: MA16-003, with an
age of 119.7 ± 0.5 ka, and MA16-004, with an age of 120.2 ± 0.6 ka.

It is unclear which coral samples were collected *in situ*/growth position, so MA16-003 and MA16-004 were not treated
as RSL indicators within the database.

### 3.16 Haiti

Bard et al. (1990) reported U-series ages from the northwest coast of Haiti. In total, three analyses were conducted on two
unique coral specimens, with one of the corals dated in duplicate. All three ages were accepted by the study authors, but
detrital Th content was not reported, which meant we were unable to accept the ages based on the strict screening protocol.
Under the flexible protocol, all three ages were accepted from two corals: BA90-021 (122.8 ± 1.1 ka) and BA90-022
(weighted mean age of 125.3 ± 1.4 ka). Both samples were identified as *Acropora palmata* corals that were part of a reef
crest facies, which typically grows in < 5 m water depth (Lighty et al., 1982). The NE coastline of Haiti is tectonically
active, and it is estimated that the uplift rate is approximately 0.36 m/kyr based on the elevation of the local LIG terrace
reported by Dodge et al. (1983).

### 3.17 Hawaii, USA

Several studies have published coral U-series ages from the Waimanalo Formation on the Hawaiian island of Oahu, which
dates to the LIG (Hearty et al., 2007; McMurtry et al., 2010; Muhs et al., 2002b; Szabo et al., 1994). Oahu is slowly uplifting
at a rate of ~ 0.06 m/kyr because it is located on the peripheral bulge of the island of Hawai'i, which is subsiding as a
consequence volcanic loading from Hawaiian hotspot volcanism (McMurtry et al., 2010 and references therein; Szabo et al.,
1994). A total of 82 U-series analyses were published for Oahu, from 72 unique coral specimens. Eight of the samples have
been dated in duplicate and one in triplicate.

In the original studies, the authors accepted 59 ages from 52 coral samples, whereas under the strict screening protocol,
we accepted 25 U-series ages from 23 coral samples. Using the flexible screening protocol, this number increased to 34 ages
from 29 coral samples, with ages ranging from 110.84 ± 3.9 ka (SZ94-007) to 133.0 ± 3.3 ka (SZ94-021). The $^{232}$Th
concentrations for Szabo et al. (1994) were recalculated using the published $^{230}$Th/$^{232}$Th activity ratios. Additionally, we
interpreted the limit of quantification for the XRD measurements in Szabo et al. (1994) to be 5 % for calcite, which led to
two additional ages (SZ94-002-001, SZ94-016-001) being accepted.

Many of the dated samples from Oahu were either clasts or collected from marine conglomerates, so the number of
samples that can be used as RSL indicators is substantially smaller than the total number of corals that passed screening.
Under the flexible protocol, a total of nine corals can be treated as RSL indicators, with ages ranging from 110.8 ± 3.9 ka
(SZ94-007) to 126.5 ± 0.7 ka (MU02-055). The only constraint on paleowater depth was given in Szabo et al. (1994), in
which the authors noted that typical water depths for Pacific *Pocillopora* and *Porites* are between 1 to 27 m based on a
previous synthesis paper (Wells 1954). Both Szabo et al. (1994) and Muhs et al. (2002b), however, ultimately treated



primary growth position corals as marine-limiting, with a minimum water depth of 1 m. Without any additional constraints on water depth, we applied the taxon-specific modern coral depth distributions of (Hibbert et al., 2016).

### 3.18 Indonesia

Coral U-series ages were reported for two locations in Indonesia, Sumba Island and Southeast Sulawesi (Bard et al., 1996;
Pedoja et al., 2018). Both locations are tectonically active, with uplift rates for Sumba Island and Southeast Sulawesi estimated by the study authors to be 0.2 – 0.5 m/kyr and 0.12 – 0.29 m/kyr, respectively. Between these two study sites, a total of 21 corals were dated, and 14 of these ages were accepted by the study authors. Based on the strict closed-system criteria, four of the ages from Pedoja et al. (2018) were accepted: PE18-001, with an age of $133.7 \pm 3.0$ ka; PE18-002, with an age of $131.2 \pm 3.0$ ka; PE18-005, with an age of $112.8 \pm 3.0$ ka; and PE18-008, with an age of $127.8 \pm 2.0$ ka. Under the
flexible criteria, six additional ages were accepted from Bard et al. (1996) ranging from $86.9 \pm 0.6$ ka (BR96-008) to $133.1 \pm 1.0$ ka (BR96-012).

Although sample elevations for Pedoja et al. (2018) were reported, the elevation uncertainty is large ($\pm 10$ m), and the authors did not provided facies information or state whether the corals were in primary growth position. All six Bard et al. (1996) ages were identified as primary growth position corals. Sample BR96-016 was originally interpreted as growing in 5-
15 m water depth, so we have used the midpoint of this range as the assigned paleowater depth uncertainty (*i.e.,* 10 +5/-5 m). The other five ages were not associated with facies/paleowater depth interpretations, so the modern taxa depth distributions of Hibbert et al. (2016) were assigned. However, it should be noted that sample BR96-017 was identified as a *Porites* microatoll, which implies that the colony was likely growing within the subtidal/intertidal zone.

### 3.19 New Caledonia

One study reported U-series ages from corals cored on Amédée Islet, New Caledonia (Frank et al., 2006). In total, 19 analyses were reported for 15 corals, with one coral dated five times. The study authors used open-system ages (Thompson et al., 2003; Villemant and Feuillet, 2003) to confirm the existence of a Last Interglacial reef deposit within the core records and estimated a subsidence rate of 0.16 m/kyr for the study site. The strict and flexible closed-system criteria rejected all of the ages from New Caledonia.

### 3.20 Niue

A single coral with a late MIS-6 age (KE12-001; $133.5 \pm 1.0$ ka) was reported for the South Pacific island of Niue and has been interpreted as a *Porites* microatoll that infilled a karstic channel (Kennedy et al., 2012). The authors interpreted this deposit as being Last Interglacial in age. This U-series age, however, has an anomalously low initial uranium isotopic value ($\delta^{234}U_i = 121.7 \pm 3.3$ ‰) and fails both the strict and flexible closed-system screening criteria.



### 3.21 Papua New Guinea

LIG fossil coral U-series ages are available from uplifted coral reef terraces on the Huon Peninsula, Papua New Guinea (Cutler et al., 2003; Esat, 1999; Stein et al., 1993). The region has experienced substantial uplift since the LIG, with local uplift rates estimated to be ~ 2 m/kyr. As a result, the LIG fossil reef deposits are presently located ~140 – 230 m amsl. A total of 47 analyses were reported from Huon Peninsula fossil reefs from 34 unique coral specimens. One coral was dated in duplicate, and 5 corals have U-series ages from multiple subsamples. Of the 47 U-series analyses performed, the study authors accepted 11 ages from 7 coral samples (although the actual number of ages accepted is likely higher, as Esat et al. (1999) did not specify the acceptable $\delta^{234}U_i$ thresholds used in their study). Under the strict screening protocol, 11 ages from 5 coral samples (CU03-011, ST93-005, ST93-006, ST93-007, ST93-009) were accepted, whereas 13 ages from 7 samples (CU03-011, CU03-023, ES99-020, ST93-005, ST93-006, ST93-007, ST93-009) were accepted under the flexible protocol. The ages that pass the flexible screening protocol range from a weighted mean age of 115.2 ± 0.7 ka for CU03-011 to 136.8 ± 1.8 ka for a single analysis from ST93-006.

Corals CU03-011, CU03-023 and ES99-020 do not have any contextual information that can be used to determine if they are in primary growth position, so we did not treat these samples as RSL indicators. ST93-005, ST93-006, ST93-007 and ST93-009 are in primary growth position, but there are no published paleowater depth interpretations provided. Additionally, there are insufficient modern observations for *Gardinoseris planulata* to produce a robust modern depth distribution for these samples, so samples ST93-005 through ST93-007 can only be treated as marine-limiting RSL indicators (Hibbert et al., 2016). The final sample, ST93-009 is a colony of *Porites lutea*, which has a modern depth range of 0 +0/-45 m (Hibbert et al., 2016).

### 3.22 Saudi Arabia

One study published coral U-series ages for emergent coral reef terraces along the Red Sea coast of Saudi Arabia (Manaa et al., 2016). In total, study authors reported U-series ages for 25 coral samples and accepted 17 of the ages. Using the strict closed-system screening criteria, that number is reduced to three samples collected from reef terraces near the port city of Yanbu. Two of the samples (MA16-009 and MA16-010) were collected from the Lower Yanbu Terrace and yielded ages of 42.2 ± 0.1 and 51.4 ± 0.1 ka, respectively, but were rejected because the ages were not stratigraphically consistent with the rest of the unit (Manaa et al., 2016). The remaining age, from the Upper Yanbu Terrace (MA16-013) yielded a Last Interglacial age of 127.9 ± 0.4 ka. A second sample from the Upper Yanbu terrace (MA16-012) was also accepted under the flexible screening protocol, yielding an age of 112.6 ± 0.3 ka. The authors did not state whether these samples were *in situ*/growth position, so these samples were not treated as RSL indicators.



### 3.23 The Seychelles

Two studies from the Seychelles published U-series coral ages (Dutton et al., 2015b; Israelson and Wohlfarth, 1999), containing a total of 67 U-series measurements for 31 individual coral specimens. Approximately half (15) of the corals were dated in triplicate, with three corals measured in duplicate. In the original studies, 24 of the corals yielded acceptable ages (38 unique U-series ages). Under the strict screening protocol, only five unique U-series ages from three corals are accepted. This is increased to 25 U-series ages from 14 corals once the flexible screening criteria is applied, with ages ranging from a

weighted mean age of $122.2 \pm 0.5$ ka from sample DU15-017 to $129.1 \pm 1.6$ ka from sample IS99-007.

Multiple sources have interpreted the Seychelles deposits as having formed in an intertidal to upper subtidal zone, which, based on modern analogues from these same islands, results in a maximum water depth of 2 m (Dutton et al., 2015b; Israelson and Wohlfarth, 1999; Montaggioni and Hoang, 1988). Here, the maximum paleowater depth of 2 m was adopted, with one exception. Sample IS99-010, a *Porites* sp., was not explicitly tied to the subtidal facies. Therefore, we assigned

IS99-010 a water depth uncertainty based on the modern depth distribution for *Porites* sp. (4 +62/-4 m; Hibbert et al., 2016). Of the 14 samples that met the flexible closed-system screening criteria, all except DU15-017 and DU15-019 were identified as being in primary growth position.

### 3.24 South Australia

Last Interglacial coral ages were reported for subtidal deposits on the Yorke Peninsula near Adelaide, South Australia (Pan

et al., 2018). Four U-series ages were reported by the authors from 4 unique specimens of the solitary coral *Plesiastrea versipora*. One of the ages (PA18-002) was originally accepted by the authors, but none of the four samples met the strict nor the flexible closed-system screening criteria. The *Plesiastrea versipora* were not in primary growth position and these deposits were interpreted by Pan et al. (2018) to be wave- and/or storm-derived.

### 3.25 St. Croix, US Virgin Islands

In the US Virgin Islands, sediment cores from Holocene reefs off the island of St. Croix possess Last Interglacial reef deposits in the underlying substrate (Toscano et al., 2012). St. Croix is unique among many other Carribean LIG sites in that only all but one of the LIG fossil reef localities is presently submerged below modern sea level, and there is an apparent 0.62 m/km gradient between LIG deposits from the northwestern to northeastern ends of the island. The authors interpreted this gradient as having resulted from differential subsidence or tilting caused by regional tectonism (Toscano et al., 2012 and

references therein). Six corals from this study yielded Last Interglacial ages, five from drill cores on Tague Reef on the northeastern end of St. Croix and one from a drill core farther west, on Long Reef. Toscano et al. (2012) accepted all ages, except the one from Long Reef (TO12-010). The strict closed-system criteria yielded similar results, but it also rejected coral TO12-008 from Tague Reef because of an elevated $\delta^{234}U_i$ value. The four corals that passed the strict screening criteria are: TO12-005, with an age of $115.1 \pm 0.3$ ka; TO12-006, with an age of $124.6 \pm 0.3$ ka; TO12-007, with an age of $123.44 \pm 0.4$



ka; and TO12-009, with an age of 129.4 ± 0.4 ka. An additional sample, TO12-008 can also be included under the flexible protocol, yielding an age of 125.7 ± 0.3 ka.

In Toscano et al. (2012), corals that were not in primary growth position were listed as "fragments." Since none of the corals that passed geochemical screening were stated as being fragments in the Toscano et al. (2012) supplement, we have treated them as being in primary growth position. The authors interpreted the Tague reef LIG deposit as being part of a reef

flat/backreef setting in <5 m water depth, so an interpreted paleowater depth range of 5 +0/-5 m was adopted. Based on the U-series dating, Toscano et al. (2012) estimated subsidence rates of 0.08 m/kyr for Tague Reef and 0.07 m/kyr for the Long Reef site.

### 3.26 Turks and Caicos

U-series ages have been reported for 19 corals collected from the LIG Boat Cove and South Reef units on West Caicos

(Kindler and Meyer 2012; Kerans et al. 2019). Additional mass spectrometric U-series ages have been reported by Simo et al. (2008), but the type of material dated ranged from "well preserved coral, to skeletal grains and ooids". As Simo et al. (2008) did not specify which carbonate material was dated for each of their ages, their dataset was not included in the present compilation.

Of all the ages reported, a total 13 were accepted by the study authors. Under both the strict and flexible screening

criteria, the two ages from Kindler and Meyer (2012) were rejected due to high calcite content. For the Kerans et al. (2019) study, calcite content was not reported for each sample, but the authors stated that XRD measurements indicated "a range of calcitization 100 % calcite to 3 % calcite", so the authors instead used Sr element mapping to identify the best preserved sections of coral to date. Based on the range of calcite concentrations given, none of the samples would pass the strict/flexible protocols, so these samples were also rejected.

### 3.27 Vanuatu

Edwards et al. (1987) reported Last Interglacial U-series ages for uplifted coral terraces on Efate Island, Vanuatu. In total, there are three U-series ages for two corals, with one coral (ED87-010) measured in duplicate. All three ages were accepted by the study authors and also passed the strict and flexible closed-system criteria. ED87-010 has a weighted mean age of 130.6 ± 1.1 ka, which constrains the age of the lower Efate terrace, and ED87-011 has an age of 126.5 ± 1.4 ka which

constrains the age of the upper terrace. No elevation information was reported for these samples, so they cannot be used as RSL indicators.

### 3.28 Western Australia

Fossil corals from the coastline of Western Australia represent perhaps the broadest geographic region reported here, spanning more than 1400 km from Cape Range in the north to Foul Bay near the southwestern tip of the Australian

continent. It has one of the largest number of U-series ages of any study area covered by WALIS, with 176 U-series ages



reported for 156 unique coral specimens (Collins et al., 2003; Eisenhauer et al., 1996; Hearty et al., 2007; Leary et al., 2013; McCulloch and Mortimer, 2008; O'Leary et al., 2008b, 2008a; Stirling et al., 1995, 1998). These sites are considered to be tectonically stable, with one notable exception being sites near the Cape Cuvier anticline, for which there is strong evidence of neotectonism since the LIG (Whitney and Hengesh, 2015).

In the original studies, at least 59 of the U-series ages from 55 coral were accepted. Under the strict screening protocol, the total number of accepted ages dropped substantially to just nine, from five corals, largely because calcite content was not reported in many of the studies. This resulted in 75 % of the dataset being summarily rejected without any assessment of age quality. To remedy this, for the flexible screening protocol, we allowed samples without reported calcite content to be screened using $^{232}$Th and $\delta^{234}U_i$. This is a reasonable judgement call to make, as in previous studies by the same authors only

corals with calcite contents below detection limit were dated (*e.g.,* Stirling et al., 1995). Under the flexible screening protocol, we also accepted ZH93-001-001, which had slightly elevated $^{232}$Th concentration (~3 ppb), and both samples from SI96-002, as they fell along the 129-ka, closed-system isochron. Samples that passed flexible screening were comparable to those that were accepted in the original publications, with 61 ages accepted from 56 unique coral samples. Ages ranged from $116.3 \pm 0.3$ ka (ST98-012) to $134.3 \pm 1.9$ ka for sample EI96-006.

Of the samples that passed the flexible screening protocol, 42 were in primary growth position (45 analyses total) and can be used as RSL indicators. Explicit paleowater depth interpretations were not provided in most cases, so we assigned the modern coral depth distributions of Hibbert et al. (2016). It should be noted, however, that many of the samples in the dataset were collected from the very top of the LIG reef outcrops and were likely growing within a few meters of sea level. For samples from Shark Bay (OL08-002; OL08-003; OL08-009; OL08-010), no coral taxonomic information was provided,

so these samples should be treated as marine-limiting RSL indicators. Additionally, two samples from the Houtman Abrolhos Islands (ZH93-001; ZH93-005) were interpreted to be intertidal or subtidal deposits, and can be constrained to < 2 m paleowater depth (Zhu et al., 1993).

### 3.29 Yemen

Fossil coral U-series ages were reported for emergent reef terraces in Yemen, along the Al-Hajaja coast and on Perim Island

(Al-Mikhlafi et al., 2018). In total, 35 U-series ages were reported for 33 coral specimens, with two corals dated in duplicate. Al-Mikhlafi et al. (2018) concluded that terrace Tr3 from the study area was Last Interglacial in age, but decided against using any of the samples collected as RSL indicators, as most of the corals were diagenetically altered. None of the U-series ages from Yemen met the strict nor flexible closed-system criteria.

### 3.30 Yucatán, Mexico

Two reef tracts were uncovered during the excavation and construction of the Xcaret theme park near Playa del Carmen, Mexico. U-series ages are available from both reef tracts, which included *Acropora palmata* and *Siderastrea siderea* corals (Blanchon et al., 2009). In total, 33 U-series ages were reported from 10 unique coral specimens, with each coral dated at

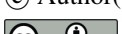



least in triplicate. In total, the study authors accepted seven analyses from five corals. Both the strict and flexible screening criteria rejected all but three analyses, primarily because multiple subsamples that were dated for the rejected coral

specimens did not yield reproducible U-series ages. The three ages that passed screening were both from upper reef tract sample BL09-006, giving a weighted mean age of $123.9 \pm 0.7$ ka. This sample, however, was identified as a clast by Blanchon et al. (2009) and therefore cannot be used as an RSL indicator.

## 4 Further details

### 4.1 Other interglacials

Multiple studies published coral U-series ages > 150 ka, suggesting the corals grew during previous glacial-interglacial cycles (*e.g.,* Andersen et al., 2010b, 2008; Bard et al., 1991; Camoin et al., 2001; Gallup et al., 1994; Hearty et al., 1999; Kennedy et al., 2012; McMurtry et al., 2010; Muhs et al., 2011; Stirling et al., 2001; Thomas et al., 2012; Vezina et al., 1999; Zazo et al., 2007). Assessing the quality of pre-LIG fossil coral U-series ages would require open-system modelling, which is beyond the scope of this study.

### 800 4.2 Holocene Coral Data

Holocene coral U-series ages are not included in this study. These data are, however, being compiled by the HOLSEA working group (Khan et al., 2019; https://www.holsea.org).

### 4.4 Controversies

One of the outstanding controversies for fossil coral RSL reconstructions is whether fossil reef sites record evidence of

millennial- or centennial- scale sea-level change within the LIG. Constraining the anatomy (pattern) of GMSL change within the LIG is crucial for our understanding of ice sheet dynamics in warm interglacial periods such as today, and has direct bearing on future projections of sea-level response to anthropogenic forcing (Church et al., 2013; DeConto and Pollard, 2016; Sweet et al., 2017). The analytical precision of U-series dating and field surveying techniques has advanced dramatically over the past 30 years, but this key question remains unresolved (Kopp et al., 2017).

### 810 5 Future research directions

Reconciling different interpretations of GMSL pathways during the LIG will require an approach that integrates age, elevation and sedimentary/facies evidence at key fossil reef sites. At the site/regional level, precise U-series age constraints are needed for key LIG fossil reef sites and must be combined with a rigorous assessment of diagenesis and its effect on U-series age quality (Dechnik et al., 2017; Dutton et al., 2015b; Obert et al., 2016; Tomiak et al., 2016). Our understanding of

LIG sea-level change will be further advanced if efforts are made to better integrate U-series ages information within the

context of coral elevation and existing site metadata (*e.g.,* facies analysis, paleoecological interpretations). Moving forward, several "best practices" that can further this goal include:

(1)    During field collection, the vertical position and depositional context should be thoroughly documented, including an assessment of whether the sampled coral is in primary growth position

(2)    Whenever possible, at least three subsamples of an LIG fossil coral should be dated to screen for open-system behavior and verify age reproducibility.

(3)    Finally, U-series ages that are accepted should be evaluated in the context of existing facies and paleoecological interpretations for the study site, to quantify the paleowater depth uncertainty for each fossil coral RSL data point.

These interpretations are needed to ensure that U-series-dated fossil corals continue to provide robust RSL information that 825   can answer important questions about LIG sea level.

## 6 Data Availability

The current version of the dataset can be accessed using the following link: http://doi.org/10.5281/zenodo.4309796 (Chutcharavan and Dutton 2020), and the descriptions of the different database fields can be found here: https://doi.org/10.5281/zenodo.3961543 (Rovere et al., 2020). The version of the coral U-series dataset referenced in this 830   manuscript is included in supplementary Tables S1 and S2.

## Author Contribution

P. Chutcharavan compiled the coral U-series database, developed the closed-system screening protocols and wrote the manuscript with assistance and guidance from A. Dutton. P. Chutcharavan and A. Dutton designed the U-series database structure.

## 835   Competing Interest

The authors declare that they have no competing interest.

## Acknowledgements

This database was compiled in WALIS, a sea-level database interface developed by ERC Starting Grant "WARMCOASTS" (ERC-StG-802414), in collaboration with the PALSEA (PAGES/INQUA) working group. The database structure was 840   designed by A. Rovere, D. Ryan, T. Lorscheid, A. Dutton, P. Chutcharavan, D. Brill, N. Jankowski, D. Mueller, and M. Bartz.



Support for the fossil coral database compilation was also provided by NSF grants #1702740 and #1443037. We would like to thank A. Rovere for his assistance in designing and implementing the U-series and fossil coral RSL components of the WALIS interface as well as Karla Rubio Sandoval, who helped compile data for the portions of the Curaçao fossil reef sites.

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
