# Peer review of "A Global Compilation of U-series Dated Fossil Coral Sea-level Indicators for the Last Interglacial Period (MIS 5e)"

_Earth System Science Data, 2020_

## Referee Comment (RC1) · Anonymous Referee #1 · 17 Jan 2021

General Comments: The manuscript is well written, well structured and of an overall appropriate length. In my eyes, it is a very useful compilation for researchers working on this topic, but it will also be of value to workers not very familiar with the details of the topic. The way the methods, results and interpretations are presented is concise, but provides sufficient details on the individual aspects. I was able to follow the reasoning of the authors and the explanations for their approach as well as the implications for future research. Compilations like this one will always be important and help make data more approachable to researchers.

Following the guidelines provided for reviewers, I think the data is of high quality, the

paper is appropriate to explain the database provided and I did not find any problems with the database itself. The length and structure of the article are appropriate, the language is consistent and precise and figures and tables are correct and of high quality. Overall, I would rate this manuscript "excellent" and think it deserves publication.

Specific Comments: The only specific comment I have is related to chapter 2.2. In lines 138 - 139, the wording could be improved. The way it is now could suggest that 238U is ingrowing, which is not the case. Maybe this could be made clearer. In addition, I feel like one or two references should be added, providing a reader with sources for more information about the method.

Technical Corrections: A short list of typos/inconsistencies: L. 90: "seeking" L. 124 - 125: It is not entirely clear to me what exactly was back-calculated. L. 158: Maybe add a reference to the respective chapter here. L. 180, l. 209: Reference to "Fig. 1" is wrong, this should be Fig. 3. L. 323: I think there is a word missing between "comparison" and "modern". L. 344: "because of GIA effects..." L. 373: not sure if I missed it, but I think the abbreviation "DT" was not defined L. 375: "topography may affect interpretations" L. 378: "GMSL", not "GSML" Throughout Chapter 3, it was sometimes inconsistent, when the authors chose to write out the numbers ("seven") or use numerals ("7"). Similarly, abbreviations (for instance "LIG") were sometimes used, sometimes not. L. 505: "Curaçao" L. 667 and l. 671: Is it "Esat (1999)" or "Esat et al. (1999)"?

---

## Short Comment (SC1) · 3 Feb 2021

I do appreciate this data compilation and the effort to update the Hibbert et al. (2016) database. I am wondering however, why Medina-Elizalde (2013; EPSL) was not referenced? Yes, it is true, when screening the existing data, Medina-Elizalde did not consider spike calibration and this may well have a significant impact on the accuracy of the ages used for inferring sea-level fluctuations. When looking at the screening protocol employed here, I do find the 'flexible' protocol only in the table and this means that age inaccuracies may also be part of the database. There is no mention on spike calibration in the text and the relevant column is empty. Expanding the limit of d234U value

to 140-152 ‰ is a prudent decision. It reflects the desire to include more (or any?) data, given the general experience that the modern d234U value of 146.8$\pm$0.1 ‰ (Andersen et al. 2010) is indeed hardly obtained. The unresolved bias in inter-laboratory comparison, as the authors put it rightly, is a key issue here. My concern here would be if age data generated on the basis of this approach are used to infer sea-level fluctuation(s) within the LIG time interval. I think the paper should make this very clear. One way to stress the implication of the flexible protocol would be to change the uncertainty of the age by adding a systematic error of 4% to the analytical error of the measurement where the 4% would reflect the d234U=146$\pm$6 ‰ used here. It is good practice in dating techniques to account for known, but hard to quantify errors associated with the dating procedure, in particular when the age uncertainty is based on counting statistics only. With the sea-level fluctuation(s) in mind I would go even further. I think a new, updated database should address the uncertainty calculation in great detail by listing systematic errors (e.g., reference material, instrumental reproducibility), counting error and other analytical errors. In this way the bias in inter-laboratory comparison would be addressed and the age estimation would approximate what we do today already when estimating the elevation, i.e., summing up all errors associated with the value.

---

## Author Comment (AC1) · 4 Feb 2021

We would like to thank Anonymous Referee #1 for the helpful feedback and comments. We agree that the statement "radioactive decay and ingrowth of 230Th, 234U and 238U" in line 139 could be misinterpreted as implying that ingrowth of 238U is occurring. This statement will be rewritten as "ingrowth of 230Th from the radioactive decay of 234U and 238U", and we will cite several papers on the U-series dating method as suggested. The technical corrections are also appreciated and will be addressed during the manuscript revision process.

[Figure]

2020.

---

## Author Comment (AC2) · 13 Feb 2021

We thank Barbara Mauz for the manuscript suggestions and comments. The reason we did not cite Medina-Elizalde (2013) in lines 48-49 is because we did not use the Medina-Elizalde (2013) database for this study. We agree that the Medina-Elizalde (2013) database is relevant to the literature review and will reference this study in Section 1. We would also like to note that the "strict" screening criteria is available in the supplementary file that accompanies this manuscript. The WALIS database does not yet have the functionality to accept multiple screening protocols, which is why only the "preferred" criteria is included in the Zenodo file.

[Figure]

Regarding the comment about spike calibration information, this is indeed included within the file uploaded to Zenodo. It is in the fields titled "Calibration method for 230Th/238U ratio" and "Calibration method for 234U/238U ratio" which are columns BK and BL in the "U-series (corals)" tab. Perhaps you are referring to columns CP and CQ: "Reference material name for 230Th/238U" and "Reference material name for 234U/238U". These columns, and columns CR-CV were included to address the potential interlaboratory spike calibration biases you referred to. The goal here is that future WALIS users will be able to input a correction factor for the measured 230Th/238U and 234U/238U activity ratios in the event that such a spike calibration bias is discovered in the future. One example of this is when a laboratory spike has been calibrated to standard that is assumed to be in secular equilibrium, such as the Harwell Uraninite (HU-1). In the case of HU-1, it was later discovered that there were systematic offsets in the U-series isotopic composition of different HU-1 aliquots. There some cases where there is sufficient information in a published manuscript text to perform an ad hoc correction for the 234U/238U activity ratio (e.g., Chutcharavan et al., 2018). However, the most robust way to perform the correction is by re-measuring the HU-1 aliquot relative to a gravimetrically-calibrated standard, which is outside the scope of the present study. We acknowledge that the spike/decay constant normalization procedure was not explicitly described in the main text and will address this during the revision process.

We agree that the screening protocols applied here do not remove every instance of systematic bias and/or diagenesis and share your concern that accepting the screened data without further evaluation could result in misleading conclusions about sea-level fluctuations within the Last Interglacial Period (LIG). Due to these potential pitfalls, it is of the utmost importance that users consider available stratigraphic and facies evidence when evaluating the quality of the U-series ages contained within this dataset, and we have explicitly stated this in the manuscript text. We did not include sea-level interpretations within the manuscript/dataset, as this was outside the scope of the WALIS special issue and Earth System Science Data.
Finally, we appreciate the suggestions for addressing systematic uncertainties within the dataset. However, we do not believe that adding a uniform systematic error to each age will accurately address the issue, as this assumes that we know the effect of an elevated d234Ui value on a sample's age and that we precisely know the d234U value of Last Interglacial seawater, which we do not. It is true that many samples from certain locations (e.g., Western Australia and Barbados) fall along the Thompson et al. (2003) open-system arrays, but this is not always the case, as shown in Fig. 1 of our preprint text. Therefore, if the d234Ui value is higher than that of the ambient seawater then it may be biased to an older age, but ultimately that depends on the style of open-system behavior. We also would like to clarify that the many of the systematic errors you mentioned are typically already incorporated into the published analytical uncertainties as part of the data reduction process. The U-series community has recently established a set of minimum data reporting standards to minimize such systemic errors (Dutton et al., 2017). We have strived to adhere to these standards when assembling this dataset.

Chutcharavan, P. M., Dutton, A. and Ellwood, M. J.: Seawater 234U/238U recorded by modern and fossil corals, Geochim. Cosmochim. Acta, 224, doi:10.1016/j.gca.2017.12.017, 2018.

Dutton, A., Rubin, K., Mclean, N., Bowring, J., Bard, E., Edwards, R. L., Henderson, G. M., Reid, M. R., Richards, D. A., Sims, K. W. W., Walker, J. D. and Yokoyama, Y.: Quaternary Geochronology Data reporting standards for publication of U-series data for geochronology and timescale assessment in the earth sciences, Quat. Geochronol., 39, 142–149, doi:10.1016/j.quageo.2017.03.001, 2017.

Medina-Elizalde, M.: A global compilation of coral sea-level benchmarks: Implications and new challenges, Earth Planet. Sci. Lett., 362, 310–318, doi:10.1016/j.epsl.2012.12.001, 2013.

Thompson, W. G., Spiegelman, M. W., Goldstein, S. L. and Speed, R. C.: An open-system model for U-series age determinations of fossil corals, Earth Planet. Sci. Lett.,
210(1–2), 365–381, doi:10.1016/S0012-821X(03)00121-3, 2003.

---

## Referee Comment (RC2) · Anonymous Referee #2 · 20 Mar 2021

**General comments**

The manuscript is well written and clearly structured. It presents helpful geochemical screening protocols of Last Interglacial (LIG; MIS 5e) corals, and therefore provides an important basis for the present MIS 5e sea-level database.

I am concerned about the overlap of the present manuscript with other manuscripts of the WALIS Special Issue addressing LIG corals as sea-level indicators in various geographic regions. The primary focus of the manuscript by Chutcharavan and Dutton is on the U-series aspect of LIG corals, but in Section 2.4 the authors also focus on growth position and paleowater depth interpretations as well as tectonics and Glacial

Isostatic Adjustment (GIA). The discussion on paleowater depth in Section 2.4 appears to be beyond the scope of the paper as other manuscripts of this Special Issue should address this topic so that it should not be part of this manuscript. The authors should have exclusively focused on the geochemical screening to avoid any overlap with other manuscripts of the WALIS Special Issue. In order to avoid redundancy and therefore not to confuse the audience, the authors should refer to the other WALIS databases and manuscripts addressing uplift/subsidence rates, paleowater depths and growth position (*in situ* vs reworked samples) of LIG corals in the respective geographic regions.

As the manuscript claims to be a global compilation of LIG sea-level indicators, the authors should explore if they did the geochemical screening for all datapoints summarized in the other articles of the WALIS Special Issue, e.g., the Western Mediterranean, Madagascar, Belize and some islands/archipelagos in the tropical Pacific Ocean.

I therefore recommend that this paper could be accepted after moderate revisions regarding the aspects detailed above and the specific comments that I listed below.

**Specific comments**

- The authors should not cite Hibbert et al. (2016) for the modern depth distribution of coral taxa, but refer to the Ocean Biogeographical Information System (OBIS) as Hibbert et al. (2016) use information from the OBIS. See lines 285, 299, 472, 515, 548, 637, 652, 683, 705 and 772.

- Lines 258-259: "Growth position is usually interpreted as expressing greater confidence than *in situ*, as it implies that the coral is in the correct growth orientation or that a clear basal attachment to the reef substrate is visible at the outcrop scale." => This introduces some uncertainty as some corals in growth position

might not be *in situ*; it is not unlikely that corals have been transported, but still look like being in growth position. I agree that there is a problem with inconsistent terminology (*in situ* and growth position) in the literature and in addition many original studies do not mention if the dated corals have been collected in growth position/*in situ*.

- Lines 395-396: how many of the samples passing the two screening protocols are *in situ*?

- Lines 570-574: Thomas et al. (2009) should not be included in the WALIS database as the dataset (13 datapoints) does not present MIS 5e.

- Lines 618-619: do you consider only mass spectrometry? If not, there are many earlier studies who dated the Waimanalo formation.

- Line 662: Sample KE12-001 is not a microatoll. According to Kennedy et al. (2012), the Last Interglacial "sample was from a 2 m-high coral head and was associated with gravel clasts of other massive corals >0.5 m in diameter".

- Lines 666-667: Probably you only consider studies providing mass spectrometry U-series ages. If so, please specify. If not, Omura et al. (1994) should be mentioned as they also dated the reef terraces on Huon Peninsula (alpha spectrometry).

**Technical corrections**
**1) General formatting issues:**
Be consistent with text justified or left-aligned.
Be consistent with capitalizing words in the headings.
Be consistent with writing numbers ≤10, e.g., lines 437, 499, 519, 525, 526, 529, 535,

562, 670, 671, 673 and 710.

**2) Further formatting issues:**
Line 17: add a comma before "2020"
Lines 26-27: merge references
Edwards et al., 1987a; Edwards et al., 1987b => Edwards et al., 1987a, b
Stirling et al., 1995; Stirling et al., 1998 => Stirling et al., 1995, 1998
Line 63: add a comma before "2020"
Lines 74-76: Correct the formatting of the list, i.e., do not capitalize the first letter of the items (2) and (3); separate the items by commas or semicolons and finish item (3) with a full stop.
Line 93: add hyphen to "sea-level history"
Line 123: "in situ" in italics
Line 172: delete "that" as it appears twice
Lines 187 and 204: Do not capitalize "detrital" as it is not at the beginning of a new sentence.
Line 203: Do not capitalize "calcite" as it is not at the beginning of a new sentence.
Line 258: full stop after quotation mark
Figure 5 caption: genera in italics
Line 324: add "to" after "comparison"
Section 2.5.3: Do not use the abbreviation "DT" without explaining it once at the beginning of this section (could be done in line 370).
Lines 385-392: Use semicolons to separate the listed items and finish item (7) with a full stop. Do not capitalize the first letter of the items.
Line 437: analyses "in" total
Line 444: comma after "In total"
Line 467: corals "in" total
Line 505: correct writing of Curaçao
Line 507: comma after "In total"

Line 561 and 574: Thomas et al., 2009; a or b?
Line 757: O'Leary et al., 2008b, 2008a => O'Leary et al., 2008a, b
Line 796: Andersen et al., 2010b, 2008 => change the order of the years
Line 803: there is no item 4.3 => correct numbering
Line 805: delete space before "scale"
Line 808: "has advanced" => "have advanced"
Lines 819: add full stop

**3) Mistakes in the database:**

- WALIS U-series ID 1292: ST93-009-001 should be Stein et al. (1993) and not Cutler et al. (2003)

- WALIS U-series ID 1313: ST93-007-003 should be Stein et al. (1993) and not Esat et al. (1999)

- WALIS U-series ID 1664: according to the sample ID OL13 the reference should be O'Leary et al. (2013) instead of Stirling et al. (2001)

- WALIS U-series ID 1763, 1764 and 1765: according to the sample ID SC78 the reference should not be Muhs et al. (2012)

- WALIS U-series ID 2086, 2087 and 2088: according to the sample ID ED87-008 the reference should be Edwards et. (1987) instead of Gallup et al. (1994)

- WALIS U-series ID 2137: according to the sample ID HA91 the reference should be Hamelin et al. (1991) instead of Thompson et al. (2003)

- WALIS U-series ID 2168: change Edwards et al. (1997) to Edwards et al. (1987)

- WALIS U-series ID 2169, 2170, 2171 and 2172: according to the sample ID GA94 the reference should be Gallup et al. (1994) instead of Edwards et al. (1997)

- WALIS U-series ID 2364: according to the sample ID SC78-004 or analysis ID SZ78-004-002 (which one is correct?) the reference should not be Muhs et al. (2012)

---

## Author Comment (AC3) · 16 Apr 2021

We thank Anonymous Referee #2 for the helpful feedback and technical comments. First, we address the general comments, which focus on possible overlap between this manuscript and the other contributions to the WALIS special issue. The objective of this manuscript was to compile age, elevation and other relevant metadata for the global fossil coral record in a consistant manner. This was done following the approach of Dutton and Lambeck (2012), Hibbert et al. (2016), Medina-Elizalde (2013) and others, which treats each U-series fossil coral specimen as a single relative sea-level (RSL) indicator. Then, the geochemical and other metadata associated with these corals

can be referred to within the context of the region-specific data description papers in the WALIS special issue. The way the WALIS database is structured, as shown in Fig. 2 in the preprint text, is that fossil coral U-series analyses are uploaded into the geochronology section of the database, and then samples that can be used as RSL indicators (i.e., are in primary growth position and have an associated U-series age and elevation) are also included in the "RSL datapoint from single coral" section, which is similar to the approach employed for speleothem data (Dumitru et al., 2020). This is in contrast to the other contributions to the special issue, which focused on compiling data for a particular region.

While we have included example screening criteria to assess the quality of U-series analyses presented in this compilation, we have also emphasized that the screening applied here is only intended to be a first pass to identify clearly altered samples. The inherent meaning of a fossil coral U-series age is inseparable from the existing geologic/sedimentary evidence, and this additional context may, in many cases, necessitate the modification or outright rejection of these preliminary age interpretations. This level of interpretation was outside the scope of the WALIS special issue, but we believe the discussion of paleowater depth considerations, in situ vs growth position corals, and tectonics/GIA is essential. To provide general guidance on the considerations that should be taken into account when interpreting these data within a regional context. Otherwise, we fear the reader will accept the screened data uncritically and assume they are correct without considering these other factors (e.g., "connecting the dots" between coral RSL datapoints to infer global mean sea level without consideration of paleowater depth uncertainty and/or the effects of GIA). We have kept this discussion fairly general to guide the reader in terms of what types of issues need to be considered to avoid unnecessary overlap with other contributions to the WALIS special issue.

We wish to emphasize that the fossil coral U-series database presented here has been cross-checked at multiple stages, both during the present review process for ESSD and through additional peer review for previous versions (i.e., subsets) of the database

(Dutton et al., 2012; Hibbert et al., 2016). For the present WALIS special issue, contributions were submitted concurrently, so it was not always possible to cross-check the global compilation presented here with the other, region-specific contributions. However, one of the advantages of WALIS is that it has been designed as a "living database" that can be continuously updated and maintained after the initial release. We agree with Referee #2 that additional cross-referencing with the region-specific contributions would be beneficial (e.g., incorporating mass spectrometric U-series ages that were missed in the initial compilation) but posit that this could instead be done as an update after all submissions are received.

We also acknowledge and appreciate the substantial time and effort taken by Referee #2 to provide specific comments/technical corrections related to this manuscript. We broadly agree with the Referee's feedback and will incorporate their comments as suggested with the limited exception of the following comments outlined below:

SPECIFIC COMMENTS

1. We will cite OBIS when referring to the modern coral depth distributions, as suggested

2. Referee #2 brings up an important point that corals can be transported and yet still appear to be in situ/growth position. Indeed, we have incorporated any additional information contained within the original source manuscripts to verify whether a coral can be treated as primary growth position. For example, in describing the fossil coral dataset from the Bahamas (lines 420-431), we treated corals that were originally reported as "in situ" as not in primary growth position, as they were derived from coral rubble deposits. Similarly, There are three samples from Stein et al. (1993) collected in Papua New Guinea (WALIS U-series IDs 1318, 1319 and 1332) that are reported as being in "growth position" but are ultimately derived from detached limestone blocks and, therefore, were not treated as primary growth position. We agree that, in some cases, this additional context is not provided in the published literature. However, an a

posteriori facies assessment of the outcrops these samples were collected from would require revisiting each of the field sites described here, which is outside the scope of this study and, indeed, the WALIS special issue. We will add wording to section 2.4 to this effect.

3. We will state which samples that passed the screening protocols are in primary growth position in lines 395-396.

4. Referee #2 is correct that the Thomas et al. (2009) datapoints are not MIS 5e. We decided to adopt a wider age range of samples ($\sim$150 – 110 ka), as (a) sea-level constraints for mid-late MIS 6 are relevant to understanding the transition between Termination II and the Last Interglacial and (b) samples that are derived from an LIG outcrop can at times display spuriously older ages due to diagenesis. The WALIS database does not explicitly exclude samples that do not fit the strict definition of MIS 5e. There is functionality within the "RSL from Stratigraphy" section of WALIS to include metadata related to sedimentary features at LIG sites that are younger/older than MIS 5e. Similarly, one of the contributions to the WALIS special issue focuses entirely on MIS substages 5a and 5c (Thompson and Creveling, 2020).

5. Regarding the comment about mass spectrometry vs alpha dates: In the introduction to Section 3 (lines 396-399) we state that fossil coral U-series ages that were measured using alpha spectrometry were not included in the current version of the fossil coral dataset. The functionality to input alpha dates, however, is present in the user interface, and some contributors have already begun adding alpha dates to WALIS. This addresses Referee #2's comments regarding:

a. Lines 618-619

b. Lines 666-667

6. We will correct the error in Line 662 (and within the database) that incorrectly identifies sample KE12-001 as a microatoll when, in fact, it is a massive coral head.

TECHNICAL CORRECTIONS

1. Regarding the comment of text justified vs left-aligned: having the first paragraph of each (sub)section not indented is a formatting requirement set by the ESSD journal. We agree with all other "General formatting issues."

2. We agree with all "Further formatting issues."

3. Regarding the "Mistakes in the database": The reason why the sample/analysis IDs do not always match the study they were published in is because some coral specimens were reanalyzed in more than one publication. This is particularly common in localities such as Barbados and the Huon Peninsula, where fossil coral specimens collected in the mid-late 20th century have been reanalyzed multiple times across several studies. In cases where this occurred, sample/analysis IDs are assigned based on the first publication where U-series ages for a particular coral specimen were reported. This ensures that the user can easily distinguish which samples came from the same coral colony, which was not always clear in earlier iterations of this database. We acknowledge that this was not explicitly stated in the manuscript text and will be sure to do so during the revision process. This explanation is relevant to the issues raised regarding WALIS U-series ID numbers 1292, 1313, 1763-1765, 2086-2088, 2137, 2168 and 2169-2172. We agree that the following errors need to be corrected:

a. WALIS U-series ID 2364: The Analysis ID should be SC78-004-002, not SZ78-004-002

b. WALIS U-series ID 1664 was incorrectly labeled as coming from Stirling et al. (2001)and was actually published in O'Leary et al. (2013).

REFERENCES

Dumitru, O. A., Polyak, V. J., Asmerom, Y., and Onac, B. P.: Last Interglacial (sensu lato, ∼130 to 75 ka) sea level history from cave deposits: a global standardized database, Earth Syst. Sci. Data Discuss. [preprint], https://doi.org/10.5194/essd-2020-

387, in review, 2020.

Dutton, A. and Lambeck, K.: Ice Volume and Sea Level During the Last Interglacial, Science, 337(6091), 216–219, doi:10.1126/science.1205749, 2012.

Hibbert, F. D., Rohling, E. J., Dutton, A., Williams, F. H., Chutcharavan, P. M., Zhao, C. and Tamisiea, M. E.: Coral indicators of past sea-level change: A global repository of U-series dated benchmarks, Quat. Sci. Rev., 145, doi:10.1016/j.quascirev.2016.04.019, 2016. O'Leary, M. J., Hearty, P. J., Thompson, W. G., Raymo, M. E., Mitrovica, J. X., and Webster, J. M.: Ice sheet collapse following a prolonged period of stable sea level during the last interglacial, Nat. Geosci., 6(9), 796–800, doi:10.1038/ngeo1890, 2013.

Medina-Elizalde, M.: A global compilation of coral sea-level benchmarks: Implications and new challenges, Earth Planet. Sci. Lett., 362, 310–318, doi:10.1016/j.epsl.2012.12.001, 2013.

Stein, M., Wasserburg, G. J., Aharon, P., Chen, J. H., Zhu, Z. R., Bloom, A. L. and Chappell, J.: TIMS U-series dating and stable isotopes of the last interglacial event in Papua New Guinea, Geochim. Cosmochim. Acta, 57(11), 2541–2554, doi:10.1016/0016-7037(93)90416-T, 1993.

Stirling, C. H., Esat, T. M., Lambeck, K., McCulloch, M. T., Blake, S. G., Lee, D. C. and Halliday, A. N.: Orbital forcing of the marine isotope stage 9 interglacial., Science, 291(5502), 290–293, doi:10.1126/science.291.5502.290, 2001.

Thompson, S. B. and Creveling, J. R.: A Global Database of Marine Isotope Stage 5a and 5c Marine Terraces and Paleoshoreline Indicators, Earth Syst. Sci. Data Discuss. [preprint], https://doi.org/10.5194/essd-2021-14, in review, 2021.

---

## Author Response (AR2)

Dear Dr. Rovere,

We have completed our responses to feedback provided by Barbara Mauz and two anonymous referees during the open discussion period for our manuscript entitled "A Global Compilation of U-series Dated Fossil Coral Sea-level Indicators for the Last Interglacial Period (MIS 5e). Detailed responses are provided below, which can be cross referenced with the word document with track changes we uploaded alongside the revised manuscript. Please note that all line numbers below, unless otherwise stated, refer to the preprint text, not the revised manuscript. I have also appended responses to your additional requested technical corrections.

**Responses to Editor Technical Corrections**
*Note: All line numbers in this section refer to the revised version of the manuscript*

**Line 590. I do not remember if Hallman et al. have sites here. Please cross-check and maybe refer to their paper also in this instance.**
*Author response:* The Hallmann et al. paper was already cited on line 594. We have added a new sentence at the beginning of this section that directs the reader to Hallmann et al. (2020) for a regional overview of LIG sea-level records from French Polynesia and have now placed the in-text citation at the end of this sentence instead to be more clear.

We agree with all other technical corrections and have made the suggested changes requested by the editor.

Note: The final version of this manuscript also corrects an error where full reference for the Masson-Delmotte (2013) IPCC chapter was missing from the "References" section.

**Responses to Referee 1**

*Referee comment #1:*
The only specific comment I have is related to chapter 2.2. In lines 138 - 139, the wording could be improved. The way it is now could suggest that $^{238}$U is ingrowing, which is not the case. Maybe this could be made clearer. In addition, I feel like one or two references should be added, providing a reader with sources for more information about the method.

> *Author response to comment #1:*
> We agree that the statement "radioactive decay and ingrowth of $^{230}$Th, $^{234}$U and $^{238}$U" in line 139 could be misinterpreted as implying that ingrowth of $^{238}$U is occurring. This statement will be rewritten as "ingrowth of $^{230}$Th from the radioactive decay of $^{234}$U and $^{238}$U", and we will cite several papers on the U-series dating method as suggested.
>
> *Changes in manuscript:*
> The sentence in lines 138-139 now reads "Once the coral skeleton has formed, the U-series radiometric clock is effectively started, and the elapsed time is measured by the ingrowth of $^{230}$Th from the radioactive decay of $^{234}$U and $^{238}$U as the system returns to secular equilibrium (Edwards et al., 1987; 2003)."

*Referee technical corrections:*

**L. 90: "seeking"**
> *Author response:* Changed "seek" to "seeking"

**L. 124 -125: It is not entirely clear to me what exactly was back-calculated.**
> *Author response:* Have clarified that we back-calculated the U-series activity ratios.

**L. 158: Maybe add a reference to the respective chapter here.**
> *Author response:* We have now referenced Section 2.3.

**L. 180, l. 209: Reference to "Fig. 1" is wrong, this should be Fig. 3.**
> *Author response:* This error has been corrected.

**L. 323: I think there is a word missing between "comparison" and "modern".**
> *Author response:* added the word "to" between "comparison" and "modern"

**L. 344: "because of GIA effects..."**
> *Author response:* The missing word "of" has been added.

**L. 373: not sure if I missed it, but I think the abbreviation "DT" was not defined**
> *Author response:* We have renamed all occurrences of "DT" (lines 373, 374, 377 and 379) to "dynamic topography".

**L. 375: "topography may affect interpretations"**
> *Author response:* This error has been corrected.

**L. 378: "GMSL", not "GSML"**
> *Author response:* This error has been corrected.

**Throughout Chapter 3, it was sometimes inconsistent, when the authors chose to write out the numbers ("seven") or use numerals ("7"). Similarly, abbreviations (for instance "LIG") were sometimes used, sometimes not.**
> *Author response:* We have cross-checked Chapter 3 for the above formatting irregularities and made edits as needed. In general, numerals are use when referring to elevations or water depths (*e.g.*, "3 to 4 m" in lines 424-425), and the numbers are written out when referring to a quantity less than 10 (*e.g.*, "A total of five corals…" in line 437).
>
> *Changes in manuscript:* the following changes have been made
> - Numbers less than ten in lines 437, 467, 519, 525, 526, 562, 670, 671, 673 and 710 have now been written out instead of being entered as numerals.
> - Occurrences of the phrase "Last Interglacial" in lines 171, 382, 490, 517, 560, 577, 593, 594, 657, 663, 690-691, 709, 715, 720, 746 and 781 have been changed to "LIG".

**L. 505: "Curaçao"**
> *Author response:* This error has been corrected.

**L. 667 and l. 671: Is it "Esat (1999)" or "Esat et al. (1999)"?**
> *Author Reponse:* It is Esat et al. (1999). This has been corrected.

**Reponses to Referee 2**

*Referee comment #1:*
I am concerned about the overlap of the present manuscript with other manuscripts of the WALIS Special Issue addressing LIG corals as sea-level indicators in various geographic regions. The primary focus of the manuscript by Chutcharavan and Dutton is on the U-series aspect of LIG corals, but in Section 2.4 the authors also focus on growth position and paleowater depth interpretations as well as tectonics and

Glacial Isostatic Adjustment (GIA). The discussion on paleowater depth in Section 2.4 appears to be beyond the scope of the paper as other manuscripts of this Special Issue should address this topic so that it should not be part of this manuscript. The authors should have exclusively focused on the geochemical screening to avoid any overlap with other manuscripts of the WALIS Special Issue. In order to avoid redundancy and therefore not to confuse the audience, the authors should refer to the other WALIS databases and manuscripts addressing uplift/subsidence rates, paleowater depths and growth position (in situ vs reworked samples) of LIG corals in the respective geographic regions.

*Author response to comment #1:*
The objective of this manuscript was to compile age, elevation and other relevant metadata for the global fossil coral record in a consistent manner. This was done following the approach of Dutton and Lambeck (2012), Hibbert et al. (2016), Medina-Elizalde (2013) and others, which treats each U-series fossil coral specimen as a single relative sea-level (RSL) indicator. Then, the geochemical and other metadata associated with these corals can be referred to within the context of the region-specific data description papers in the WALIS special issue. The way the WALIS database is structured, as shown in Fig. 2 in the preprint text, is that fossil coral U-series analyses are uploaded into the geochronology section of the database, and then samples that can be used as RSL indicators (*i.e.*, are in primary growth position and have an associated U-series age and elevation) are also included in the "RSL datapoint from single coral" section, which is similar to the approach employed for speleothem data (Dumitru et al., 2020). This is in contrast to the other contributions to the special issue, which focused on compiling data for a particular region.

While we have included example screening criteria to assess the quality of U-series analyses presented in this compilation, we have also emphasized that the screening applied here is only intended to be a first pass to identify clearly altered samples. The inherent meaning of a fossil coral U-series age is inseparable from the existing geologic/sedimentary evidence, and this additional context may, in many cases, necessitate the modification or outright rejection of these preliminary age interpretations. This level of interpretation was outside the scope of the WALIS special issue, but we believe the discussion of paleowater depth considerations, in situ vs growth position corals, and tectonics/GIA is essential. To provide general guidance on the considerations that should be taken into account when interpreting these data within a regional context. Otherwise, we fear the reader will accept the screened data uncritically and assume they are correct without considering these other factors (*e.g.*, "connecting the dots" between coral RSL datapoints to infer global mean sea level without consideration of paleowater depth uncertainty and/or the effects of GIA). We have kept this discussion fairly general to guide the reader in terms of what types of issues need to be considered to avoid unnecessary overlap with other contributions to the WALIS special issue.

*Referee comment #2:*
As the manuscript claims to be a global compilation of LIG sea-level indicators, the authors should explore if they did the geochemical screening for all data points summarized in the other articles of the WALIS Special Issue, e.g., the Western Mediterranean, Madagascar, Belize and some islands/archipelagos in the tropical Pacific Ocean.

*Author response:*
We wish to emphasize that the fossil coral U-series database presented here has been cross-checked at multiple stages, both during the present review process for ESSD and through additional peer review for previous versions (*i.e.*, subsets) of the database (Dutton et al., 2012; Hibbert et al., 2016). For the present WALIS special issue, contributions were submitted concurrently, so it was not always possible to cross-check the global compilation presented here

with the other, region-specific contributions. However, one of the advantages of WALIS is that it has been designed as a "living database" that can be continuously updated and maintained after the initial release. We agree with Referee #2 that additional cross-referencing with the region-specific contributions would be beneficial (*e.g.*, incorporating mass spectrometric U-series ages that were missed in the initial compilation) but posit that this could instead be done as an update after all submissions are received.

*Referee comment #3:*
The authors should not cite Hibbert et al. (2016) for the modern depth distribution of coral taxa, but refer to the Ocean Biogeographical Information System (OBIS) as Hibbert et al. (2016) use information from the OBIS. See lines 285, 299, 472, 515, 548, 637, 652, 683, 705 and 772.

*Author response:*
We will cite OBIS when referring to the modern coral depth distributions, as suggested.

*Changes in manuscript:*

- Line 285: Removed OBIS reference, as the approach of using modern coral depth distributions for parameterizing paleowater depth uncertainty was employed by Hibbert et al. (2016).

- Lines 290, 299, 321, 472, 515, 548, 637, 652, 683, 705, 772

- For line 681, both Hibbert et al. (2016) and OBIS are referenced, as the data was originally derived from OBIS, but Hibbert et al. (2016) ultimately determined that there were insufficient modern observations to produce a statistically robust modern depth distribution for *Gardinoseris planulata*.

- OBIS has been added to the references (in the preprint, OBIS was included as an in-text citation but the full citation was accidentally not included in the references section)

*Referee comment #4:*
Lines 258-259: "Growth position is usually interpreted as expressing greater confidence than in situ, as it implies that the coral is in the correct growth orientation or that a clear basal attachment to the reef substrate is visible at the outcrop scale." => This introduces some uncertainty as some corals in growth position might not be in situ; it is not unlikely that corals have been transported, but still look like being in growth position. I agree that there is a problem with inconsistent terminology (in situ and growth position) in the literature and in addition many original studies do not mention if the dated corals have been collected in growth position/in situ.

*Author response:*
Referee #2 brings up an important point that corals can be transported and yet still appear to be in situ/growth position. Indeed, we have incorporated any additional information contained within the original source manuscripts to verify whether a coral can be treated as primary growth position. For example, in describing the fossil coral dataset from the Bahamas (lines 420-431), we treated corals that were originally reported as "in situ" as not in primary growth position, as they were derived from coral rubble deposits. Similarly, There are three samples from Stein et al. (1993) collected in Papua New Guinea (WALIS U-series IDs 1318, 1319 and 1332) that are reported as being in "growth position" but are ultimately derived from detached limestone blocks and, therefore, were not treated as primary growth position. We agree that, in some cases, this additional context is not provided in the published literature. However, an *a posteriori* facies assessment of the outcrops these samples were collected from would require revisiting each of the

field sites described here, which is outside the scope of this study and, indeed, the WALIS special issue. We will add wording to section 2.4 to this effect.

*Changes in manuscript:*

Additional wording has been added at line 268 to clarify potential caveats that exist when determining if a coral is in primary growth position. We emphasize that future fieldwork efforts to systematically characterize key LIG fossil reef sites would be helpful, albeit outside the scope of the present study.

*Referee comment #5:*
Lines 395-396: how many of the samples passing the two screening protocols are in situ?

*Author Response:*
Of the 1312 U-series analyses reported in this manuscript, 444 analyses from 330 coral colonies are RSL indicators, whereas 15 analyses from 13 colonies are marine limiting (due to either (a) lacking coral taxonomic information, or (b) that paleoecological/assemblage interpretations are unavailable and the number of observations from the Ocean Biodiversity Information System (OBIS) is too small to produce a statistically meaningful modern depth distribution). Of the samples that were treated as RSL indicators, 59 ages were accepted from 39 coral samples under the strict protocol, whereas 150 ages from 112 coral samples were accepted under the flexible protocol. Finally, for the marine limiting samples, four ages from three coral samples were accepted under the strict protocol, whereas five analyses from four coral samples were accepted under the flexible protocol.

*Changes in manuscript:*
We have inserted wording to this effect at lines 10 and 396.

*Referee comment #6:*
Lines 570-574: Thomas et al. (2009) should not be included in the WALIS database as the dataset (13 datapoints) does not present MIS 5e.

*Author response:*
Referee #2 is correct that the Thomas et al. (2009) datapoints are not MIS 5e. We decided to adopt a wider age range of samples (~150 – 110 ka), as (a) sea-level constraints for mid-late MIS 6 are relevant to understanding the transition between Termination II and the Last Interglacial and (b) samples that are derived from an LIG outcrop can at times display spuriously older ages due to diagenesis. The WALIS database does not explicitly exclude samples that do not fit the strict definition of MIS 5e. There is functionality within the "RSL from Stratigraphy" section of WALIS to include metadata related to sedimentary features at LIG sites that are younger/older than MIS 5e. Similarly, one of the contributions to the WALIS Special Issue focuses entirely on MIS substages 5a and 5c (Thompson and Creveling, 2020).

*Referee comment #7:*
Lines 618-619: do you consider only mass spectrometry? If not, there are many earlier studies who dated the Waimanalo formation.

*Author response:*
In the introduction to Section 3 (lines 396-399) we state that fossil coral U-series ages that were measured using alpha spectrometry were not included in the current version of the fossil coral dataset. The functionality to input alpha dates, however, is present in the user interface, and some contributors have already begun adding alpha dates to WALIS.

*Referee comment #8:*
Line 662: Sample KE12-001 is not a microatoll. According to Kennedy et al. (2012), the Last Interglacial "sample was from a 2 m-high coral head and was associated with gravel clasts of other massive corals >0.5 m in diameter".

> *Author response:*
> We will correct the error in Line 662 (and within the database) that incorrectly identifies sample KE12-001 as a microatoll when, in fact, it is a massive coral head.
>
> *Changes in manuscript*:
> The phrase in Line 662 now reads: "…a 2 m-high *Porites* colony that infilled a karstic channel…"

*Referee comment #9:*
Lines 666-667: Probably you only consider studies providing mass spectrometry U-series ages. If so, please specify. If not, Omura et al. (1994) should be mentioned as they also dated the reef terraces on Huon Peninsula (alpha spectrometry).

> *Author response:*
> See response to Referee comment #7.

*Technical corrections*

**Be consistent with text justified or left-aligned.**

> *Author response:* Having the first paragraph of each (sub)section not indented is a formatting requirement set by the ESSD journal.

**Be consistent with capitalizing words in the headings.**

> *Author response:* Only the first word, proper nouns and acronyms in each heading are now capitalized.

**Be consistent with writing numbers _10, e.g., lines 437, 499, 519, 525, 526, 529, 535, 562, 670, 671, 673 and 710.**

> *Author response:* This has been addressed in the response to Referee #1.

**Line 17: add a comma before "2020"**

> *Author response:* This has been corrected.

**Lines 26-27: merge references**
**Edwards et al., 1987a; Edwards et al., 1987b => Edwards et al., 1987a, b**
**Stirling et al., 1995; Stirling et al., 1998 => Stirling et al., 1995, 1998**

> *Author response:* This has been corrected.

**Line 63: add a comma before "2020"**

> *Author response:* This has been corrected.

**Lines 74-76: Correct the formatting of the list, i.e., do not capitalize the first letter of the items (2) and (3); separate the items by commas or semicolons and finish item (3) with a full stop.**

> *Author response:* This has been corrected.

**Line 93: add hyphen to "sea-level history"**

*Author response:* This has been corrected.

**Line 123: "in situ" in italics**
*Author response:* This has been corrected.

**Line 172: delete "that" as it appears twice**
*Author response:* That has been corrected.

**Lines 187 and 204: Do not capitalize "detrital" as it is not at the beginning of a new sentence.**
*Author response:* This has been corrected.

**Line 203: Do not capitalize "calcite" as it is not at the beginning of a new sentence.**
*Author response:* This has been corrected.

**Line 258: full stop after quotation mark**
*Author response:* This has been corrected.

**Figure 5 caption: genera in italics**
*Author response:* This has been corrected.

**Line 324: add "to" after "comparison"**
*Author response:* This has already been addressed in the responses to Referee #1.

**Section 2.5.3: Do not use the abbreviation "DT" without explaining it once at the beginning of this section (could be done in line 370).**
*Author response:* This has already been addressed in the responses to Referee #1.

**Lines 385-392: Use semicolons to separate the listed items and finish item (7) with a full stop. Do not capitalize the first letter of the items.**
*Author response:* This has been corrected.

**Line 437: analyses "in" total**
*Author response:* This has been corrected.

**Line 444: comma after "In total"**
*Author response:* This has been corrected.

**Line 467: corals "in" total**
*Author response:* This has been corrected.

**Line 505: correct writing of Curaçao**
*Author response:* This has already been addressed in the response to Referee #1.

**Line 507: comma after "In total"**
*Author response:* This has been corrected.

**Line 561 and 574: Thomas et al., 2009; a or b?**
*Author response:* Neither. There is an error in my reference management software that flags the main article and supplement as two separate papers. This has been corrected within the manuscript.

**Line 757: O'Leary et al., 2008b, 2008a => O'Leary et al., 2008a, b**
    *Author response:* This has been corrected, and 2008a, b have also been merged with the O'Leary et al., 2013 reference.

**Line 796: Andersen et al., 2010b, 2008 => change the order of the years**
    *Author response:* This has been corrected.

**Line 803: there is no item 4.3 => correct numbering**
    *Author response:* This has been corrected.

**Line 805: delete space before "scale"**
    *Author response:* This has been corrected.

**Line 808: "has advanced" => "have advanced"**
    *Author response:* This has been corrected.

**Lines 819: add full stop**
    *Author response:* This has been corrected.

**Mistakes in the database**
    *Author response:* The reason why the sample/analysis IDs do not always match the study they were published in is because some coral specimens were reanalyzed in more than one publication. This is particularly common in localities such as Barbados and the Huon Peninsula, where fossil coral specimens collected in the mid-late 20th century have been reanalyzed multiple times across several studies. In cases where this occurred, sample/analysis IDs are assigned based on the first publication where U-series ages for a particular coral specimen were reported. This ensures that the user can easily distinguish which samples came from the same coral colony, which was not always clear in earlier iterations of this database. We acknowledge that this was not explicitly stated in the manuscript text and will be sure to do so during the revision process. This explanation is relevant to the issues raised regarding WALIS U-series ID numbers 1292, 1313, 1763-1765, 2086-2088, 2137, 2168 and 2169-2172. The following errors were corrected in the database:

        ▪  WALIS U-series ID 2364: The Analysis ID should be SC78-004-002, not SZ78-004-002

        ▪  WALIS U-series ID 1664 was incorrectly labeled as coming from Stirling et al. (2001) and was actually published in O'Leary et al. (2013).

    *Changes in manuscript*: Text has been inserted at line 113 to clarify this.

**Responses to short comment by Barbara Mauz:**

*Short comment #1:*
I do appreciate this data compilation and the effort to update the Hibbert et al. (2016) database. I am wondering however, why Medina-Elizalde (2013; EPSL) was not referenced? Yes, it is true, when screening the existing data, Medina-Elizalde did not consider spike calibration and this may well have a significant impact on the accuracy of the ages used for inferring sea-level fluctuations.

    *Author response:*
    The reason we did not cite Medina-Elizalde (2013) in lines 48-49 is because we did not use the Medina-Elizalde (2013) database for this study. We agree that the Medina-Elizalde (2013) database is relevant to the literature review and will reference this study in Section 1.

*Changes in manuscript:*
Medina-Elizalde (2013) has been cited as part of the text inserted at line 51 in response to short comment #3.

*Short comment #2:*
When looking at the screening protocol employed here, I do find the 'flexible' protocol only in the table and this means that age inaccuracies may also be part of the database.

*Author response:*
The "strict" screening criteria is available in the supplementary file that accompanies this manuscript. The WALIS database does not yet have the functionality to accept multiple screening protocols, which is why only the "preferred" criteria is included in the Zenodo file.

*Short comment #3:*
There is no mention on spike calibration in the text and the relevant column is empty.

*Author response:*
Regarding the comment about spike calibration information, this is indeed included within the file uploaded to Zenodo. It is in the fields titled "Calibration method for $^{230}$Th/$^{238}$U ratio" and "Calibration method for $^{234}$U/$^{238}$U ratio" which are columns BK and BL in the "U-series (corals)" tab. Perhaps you are referring to columns CP and CQ: "Reference material name for $^{230}$Th/$^{238}$U" and "Reference material name for $^{234}$U/$^{238}$U". These columns, and columns CR-CV were included to address the potential interlaboratory spike calibration biases you referred to. The goal here is that future WALIS users will be able to input a correction factor for the measured $^{230}$Th/$^{238}$U and $^{234}$U/$^{238}$U activity ratios in the event that such a spike calibration bias is discovered in the future. One example of this is when a laboratory spike has been calibrated to standard that is assumed to be in secular equilibrium, such as the Harwell Uraninite (HU-1). In the case of HU-1, it was later discovered that there were systematic offsets in the U-series isotopic composition of different HU-1 aliquots. There some cases where there is sufficient information in a published manuscript text to perform an *ad hoc* correction for the $^{234}$U/$^{238}$U activity ratio (*e.g.*, Chutcharavan et al., 2018). However, the most robust way to perform the correction is by re-measuring the HU-1 aliquot relative to a gravimetrically-calibrated standard, which is outside the scope of the present study. We acknowledge that the spike/decay constant normalization procedure was not explicitly described in the main text and will address this during the revision process.

*Changes in manuscript:*
Several sentences have been inserted at line 51 that explicitly describe the need to normalize to the same set of decay constants for $^{234}$U and $^{230}$Th and to account for systematic biases due to differences in interlaboratory spike calibration techniques. This is also now similarly stated with text inserted at line 127.

*Short comment #4:*
The unresolved bias in inter-laboratory comparison, as the authors put it rightly, is a key issue here. My concern here would be if age data generated on the basis of this approach are used to infer sea-level fluctuation(s) within the LIG time interval. I think the paper should make this very clear.

*Author response:*
We agree that the screening protocols applied here do not remove every instance of systematic bias and/or diagenesis and share your concern that accepting the screened data without further evaluation could result in misleading conclusions about sea-level fluctuations within the Last Interglacial Period (LIG). Due to these potential pitfalls, it is of the utmost importance that users

consider available stratigraphic and facies evidence when evaluating the quality of the U-series ages contained within this dataset, and we have explicitly stated this in the manuscript text. We did not include sea-level interpretations within the manuscript/dataset, as this was outside the scope of the WALIS special issue and Earth System Science Data.

*Short comment #5:*
One way to stress the implication of the flexible protocol would be to change the uncertainty of the age by adding a systematic error of 4% to the analytical error of the measurement where the 4% would reflect the d234U=146_6 ‰ used here. It is good practice in dating techniques to account for known, but hard to quantify errors associated with the dating procedure, in particular when the age uncertainty is based on counting statistics only. With the sea-level fluctuation(s) in mind I would go even further. I think a new, updated database should address the uncertainty calculation in great detail by listing systematic errors (e.g., reference material, instrumental reproducibility), counting error and other analytical errors. In this way the bias in inter-laboratory comparison would be addressed and the age estimation would approximate what we do today already when estimating the elevation, i.e., summing up all errors associated with the value.

> *Author response:*
> We appreciate the suggestions for addressing systematic uncertainties within the dataset. However, we do not believe that adding a uniform systematic error to each age will accurately address the issue, as this assumes that we know the effect of an elevated $\delta^{234}U_i$ value on a sample's age and that we precisely know the $\delta^{234}U$ value of Last Interglacial seawater, which we do not. It is true that many samples from certain locations (*e.g.*, Western Australia and Barbados) fall along the Thompson et al. (2003) open-system arrays, but this is not always the case, as shown in Fig. 1 of our preprint text. Therefore, if the $\delta^{234}U_i$ value is higher than that of the ambient seawater then it may be biased to an older age, but ultimately that depends on the style of open-system behavior. We also would like to clarify that the many of the systematic errors you mentioned are typically already incorporated into the published analytical uncertainties as part of the data reduction process. The U-series community has recently established a set of minimum data reporting standards to minimize such systematic errors (Dutton et al., 2017). We have strived to adhere to these standards when assembling this dataset.

**Other changes/edits**

**Line 33:** defined the term "ka", which had not been defined in the preprint.

**Line 115:** inserted a statement clarifying that the elevation of fossil coral samples relative to mean sea level and mean lower low water/mean low water springs was calculated using the IMCalc software package of Lorscheid and Rovere (2019), when a proximal tide gauge datum was not available

**Line 140:** fixed an error in the text between the hyphenated words "high-precision" that was causing formatting problems.

**Line 392:** Added an additional entry to the list stating that we have cited other contributions to the WALIS special issue that have discussed the U-series fossil coral ages compiled for this study.

**Line 467:** Adjusted number of analyses/corals accepted under the flexible screening protocol to account for one sample (U-series ID 2108) that is now being treated as marine limiting. This also applies to edits to lines 469 and 470.

**Line 483**: Directed readers to Thompson and Creveling (2021) for additional information about the California MIS 5 terraces

**Line 505:** Directed readers to Rubio-Sandoval et al., 2021 for regional overview

**Line 556:** Directed readers to Hallmann et al., 2020 for regional overview

**Line 619:** Directed readers to Hallmann et al., 2020 for regional overview

**Line 640:** Directed readers to Maxwell et al., 2021 for regional overview

**Line 655:** Directed readers to Hallmann et al., 2020 for regional overview

**Line 662:** Directed readers to Hallmann et al., 2020 for regional overview

**Line 667:** Directed readers to Hallmann et al., 2020 for regional overview

**Line 681:** Deleted phrase "RSL indicators"

**Line 751:** Directed readers to Hallmann et al., 2020 for regional overview

**Line 775:** Deleted phrase "RSL indicators"

**Line 786:** Directed readers to Simms 2021 for regional overview

**Line 841:** We added an acknowledgement to B. Mauz and the two anonymous referees for the helpful feedback they provided during the open discussion period.

**Line 880:** Added Boyden et al. 2021 reference

**Line 953**: Corrected error with Edwards et al. 1997 reference

**Line 962**: Corrected error with Esat et al. 1999 reference

**Line 977:** Inserted Hallmann et al. 2020 reference

**Line 1009:** Corrected error where O'Leary et al., 2013 reference was mislabeled as "Leary" and moved to correct location in reference list.

**Line 1014:** Added Lorscheid and Rovere 2019 reference

**Line 1019:** Added Maxwell et al. 2021 reference

**Line 1088:** Inserted Simms et al. 2021 reference

**Line 1123:** Inserted Thomson and Creveling 2021 reference